# Improved calcium sensor GCaMP-X overcomes the calcium channel perturbations induced by the calmodulin in GCaMP

Yaxiong Yang [1,2,3,4], Nan Liu[1,7], Yuanyuan He[1], Yuxia Liu[1], Lin Ge[1], Linzhi Zou[5], Sen Song[1,4], Wei Xiong[4,5] & Xiaodong Liu [1,2,3,4,5,6]

GCaMP, one popular type of genetically-encoded $Ca^{2+}$ indicator, has been associated with various side-effects. Here we unveil the intrinsic problem prevailing over different versions and applications, showing that GCaMP containing CaM (calmodulin) interferes with both gating and signaling of L-type calcium channels ($Ca_V1$). GCaMP acts as an impaired apoCaM and $Ca^{2+}/CaM$, both critical to $Ca_V1$, which disrupts $Ca^{2+}$ dynamics and gene expression. We then design and implement GCaMP-X, by incorporating an extra apoCaM-binding motif, effectively protecting $Ca_V1$-dependent excitation–transcription coupling from perturbations. GCaMP-X resolves the problems of detrimental nuclear accumulation, acute and chronic $Ca^{2+}$ dysregulation, and aberrant transcription signaling and cell morphogenesis, while still demonstrating excellent $Ca^{2+}$-sensing characteristics partly inherited from GCaMP. In summary, $CaM/Ca_V1$ gating and signaling mechanisms are elucidated for GCaMP side-effects, while allowing the development of GCaMP-X to appropriately monitor cytosolic, submembrane or nuclear $Ca^{2+}$, which is also expected to guide the future design of CaM-based molecular tools.

[1] Department of Biomedical Engineering, School of Medicine, X-Lab for Transmembrane Signaling Research, Tsinghua University, Beijing 100084, China. [2] School of Biological Science and Medical Engineering, Beihang University, Beijing 100083, China. [3] Beijing Advanced Innovation Center for Biomedical Engineering, Beihang University, Beijing 102402, China. [4] IDG/McGovern Institute for Brain Research, Tsinghua University, Beijing 100084, China. [5] School of Life Sciences, Tsinghua University, Beijing 100084, China. [6] Key Laboratory for Biomedical Engineering of Education Ministry, Zhejiang University, Hangzhou 310027, China. [7] Present address: School of Life Sciences, Yunan University, Kunming 650091, China. These authors contributed equally: Yaxiong Yang, Nan Liu, Yuanyuan He.  Correspondence and requests for materials should be addressed to X.L. (email: liu-lab@vip.163.com)

Calcium ($Ca^{2+}$) is one of the most versatile and ubiquitous messengers in biology, and diverse extracellular signals and intracellular events are encoded into spatiotemporal $Ca^{2+}$ dynamics to regulate numerous vital processes[1], such as proliferation, transcription, metabolism, exocytosis, contraction, apoptosis, etc. Consequently, enormous efforts have been devoted to develop and improve molecular tools to monitor and quantify the spatial-temporal dynamics of cell $Ca^{2+}$. Fluorescent imaging with genetically encoded $Ca^{2+}$ indicators (GECIs) has achieved rapid progress in visualizing $Ca^{2+}$ at the levels of cell populations, single cells or subcellular compartments[2]. Among them, the fluorescence-intensity and single-fluorophore based sensor GCaMP is one of the most successful and popular type of GECIs up to date, mostly owing to its capability of $Ca^{2+}$ measurement with impressive signal-noise ratio (SNR) and rapid response kinetics after continuous improvements and updates from GCaMP, GCaMP3 up to current GCaMP6 (Supplementary Fig. 1)[3–7]. GCaMP design is based on $Ca^{2+}$-binding protein calmodulin (CaM) as the sensing element, which interacts with $Ca^{2+}$/CaM-binding motif M13 from myosin light chain kinase (MLCK), and a circularly permuted green fluorescent protein (cpEGFP). $Ca^{2+}$-dependent interactions between CaM and its binding motifs switch the protonation state of fluorescent proteins, hence altering spectral properties of the chromophore[3,4,8]. However, in practice GCaMP reportedly causes unexpected and unwanted "side-effects" in multiple aspects. The major concern is about cell damages induced by GCaMP. For instance, GCaMP expression, usually in chronic terms, could impair the general health of cells and tissues, including cardiomegaly/hypertrophy in the heart of GCaMP2 transgenic mice (resembling exogenous CaM overexpression)[4,9], and cytotoxicity or death in neurons[5,10]. These side-effects seem to be closely associated with abnormal accumulation of GCaMP filling in the nuclei, often accompanied by attenuation and/or distortion of $Ca^{2+}$ dynamics as evidenced in neurons[5,10–12]. Another line of evidence, which is less obvious, comes from functional studies reporting the alterations of physiological $Ca^{2+}$ and intrinsic excitability, e.g., gain of function in firing rates was observed from hippocampal neurons of GCaM5G transgenic mice[13]. And recently, multiple lines of GCaMP6 mice are reported to exhibit major abnormalities in their brain activity[14]. According to most reports, these problems could be simply attributed to technical concerns of GCaMP probes. In fact, the actual levels of any sensors or probes based on interactions with the targeted analytes in live cells often raise concerns. The potential problems are of two-fold: an excessive amount of exogenous sensors would bring up "over-buffering" issues[15,16] whereas insufficient sensors would result into rather weak readouts in association with low SNR and loss of sensitivity[17]. However, these general aspects could not explain the above atypical side-effects particular to GCaMP, thus unable to provide any tangible and specific solution.

CaM is ubiquitous signal protein in cells acting as either apoCaM ($Ca^{2+}$-free CaM) or calcified CaM ($Ca^{2+}$/CaM). $Ca^{2+}$/CaM is widely involved in numerous $Ca^{2+}$-signaling cascades including gene transcription; and apoCaM also regulates the functions of diverse proteins including ion channels[18–21]. We suspect that the problems of GCaMP might be due to its potential perturbations of signaling networks and normal protein functions pertaining to CaM. Nevertheless, it remains unclear what is the major cause/mechanism responsible for the side-effects of GCaMP. Neither is any solution available yet to overcome these problems still existing to the newest GCaMP6[10].

We looked into the mechanistic details underlying GCaMP perturbations in neurons, and made the following discoveries. First, GCaMP interferes with the gating of L-type $Ca^{2+}$ channels ($Ca_V1$) in complex with CaM thus affecting $Ca^{2+}$ influx

and dynamics, evidently through the competition with distal carboxyl tail (DCT) of the channel and/or endogenous apoCaM pre-bound with the channel[21,22]. Second, GCaMP, often indicated by aberrant nuclear accumulation as the sign of severity, interferes with critical signaling events including cytonuclear CaM translocation and phosphorylation of transcription factor CREB (cAMP responsive element binding protein) in $Ca_V1$-dependent excitation–transcription (E–T) coupling[23,24]. To overcome these drawbacks in excitability ($Ca^{2+}$ dynamics) and signaling (gene expression), we newly developed GCaMP-X as our solution, by engineering an additional apoCaM-binding motif and an extra tag ensuring subcellular localization into conventional GCaMP.

In summary, we unveil that GCaMP containing CaM interferes with gating and signaling of L-type calcium channels, which disrupts $Ca^{2+}$ dynamics and gene expression. Based on these mechanistic insights, we propose, implement and validate new GCaMP-X with its CaM being protected, which not only improves the applicability, robustness and precision of GCaMP currently being distributed and utilized, but also provides important guidance for future development of GCaMP and other CaM-based sensors or actuators.

## Results

**GCaMP causes side-effects in neurons.** According to previous reports[5,7,10,25], GCaMP sensors could abnormally accumulate in the nucleus of neurons and cause multiple side-effects such as cell damages or death, altered neural functions and impaired sensing performance, more often evidenced from high levels or extended time of GCaMP expression with plasmid transfection or virus infection. Such phenomena were replicated in our experimental settings with cortical neurons infected with AAV-syn-GCaMP6f virus (Fig. 1a). In contrast to stable EGFP control, GCaMP6f exhibited a time-dependent trend of accumulation in the nucleus, which started with N/C (nuclear/cytosolic) ratio of 0.29 ± 0.02 ($n = 16$) in half a week, but gradually reaching 0.99 ± 0.01 ($n = 82$) at the timepoint of 3rd week (Fig. 1b). Such aberrant nuclear accumulation of GCaMP is reportedly problematic, usually accompanied by various side-effects. Indeed, as demonstrated by the apoptosis assay (Annexin V kit), at the time of 10th day (GCaMP N/C ratio of ~0.8) robust signals of apoptosis were detected from a large fraction (>60%) of neurons infected with AAV-Syn-GCaMP6f virus (Fig. 1c), whereas much less for control neurons infected with AAV-Syn-EGFP (~20%). To test the hypothesis that nuclear accumulation of GCaMP could underlie its side-effects in neurons, we examined and quantified the cytonuclear distribution of GCaMP in relation to neurite outgrowth. For cultured cortical neurons (5 days in vitro or DIV 5) transiently transfected with GCaMP3 cDNA, in two days after transfection, GCaMP3 distributions can be segregated into two extreme patterns by the criteria of N/C ratio of GCaMP3 fluorescence: nuclear-excluded (N/C ratio < 0.6, ~50% out of the total number of neurons) or nuclear-filled (N/C ratio > 1, ~10% of neurons) (Fig. 1d, e). Additional experiments with TagRFP-GCaMP3 confirmed that GCaMP3 fluorescent images indicative of basal $Ca^{2+}$ in cells can very well represent the cytonuclear distribution of proteins (Supplementary Fig. 2). We then examined the potential neural damages due to nuclear accumulation of GCaMP, by tracing the neurite morphology indicated by green fluorescence in each neuron with the aid of wide-field confocal microscopy when necessary (Fig. 1d, f). Nuclear-filled group was also mimicked by NLS-GCaMP3-NLS constitutively expressing in the nucleus directed by the NLS (nuclear localization signal) tag. N/C ratio of GCaMP3 and the total length of neurites were calculated and correlated for individual neurons (Fig. 1g). As

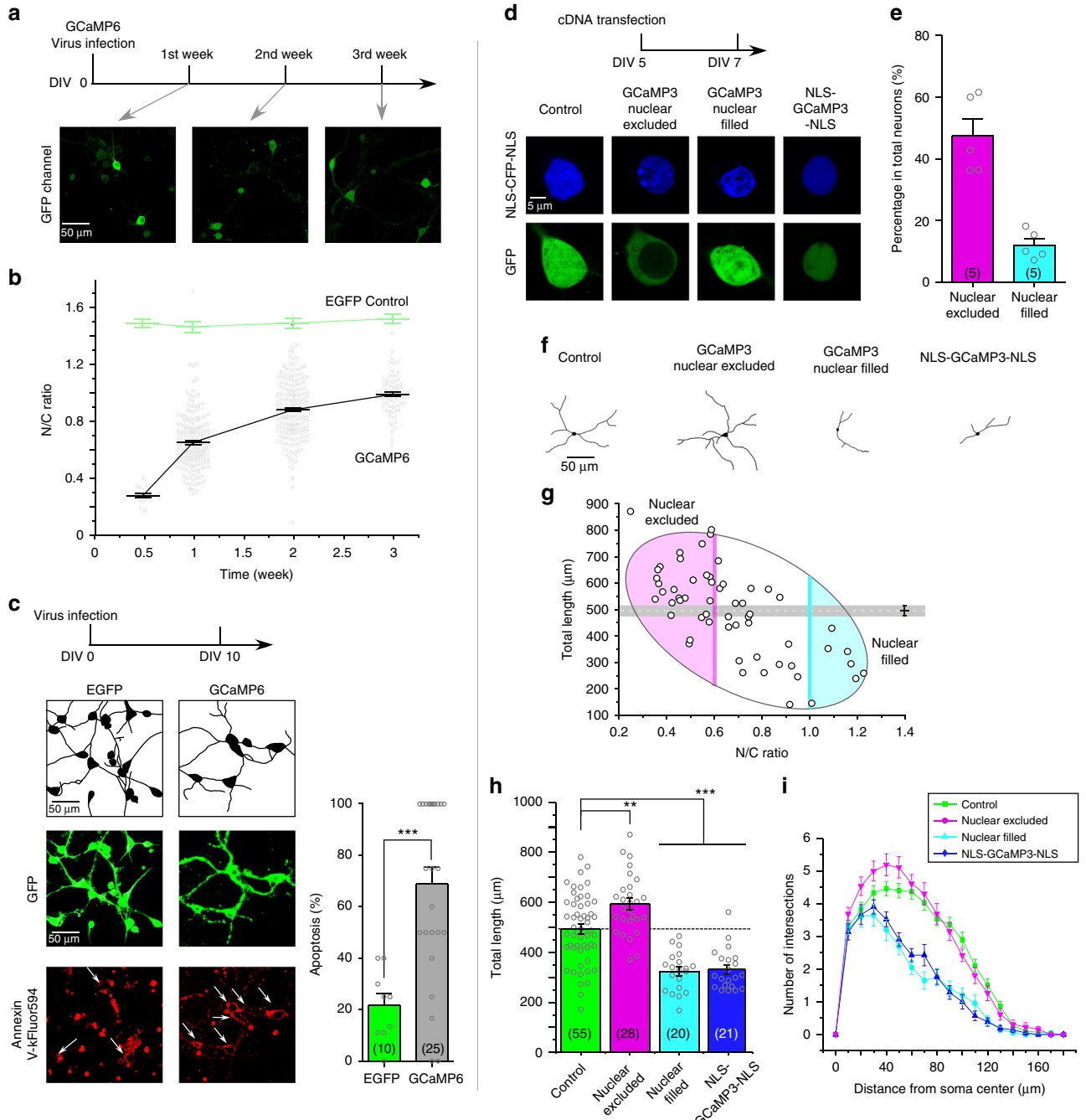

**Fig. 1** Side-effects of GCaMP on cortical neurons. **a** Abnormal nuclear accumulation of GCaMP. Cultured cortical neurons infected with GCaMP6 (AAV-*Syn*-GCaMP6f) virus were examined with confocal live-cell imaging. **b** Analyses indexed with nuclear/cytosolic (N/C) fluorescence ratio indicate time-dependent nuclear accumulation of GCaMP6, in contrast to stable GFP distribution. **c** GCaMP6 caused apoptosis. For cortical neurons infected with AAV-*Syn*-EGFP and AAV-*Syn*-GCaMP6f (green), tracing images of dendritic morphology (black and white) and confocal fluorescence images of Annexin V-kFluor594 (red) are shown. Arrows are to identify apoptotic neurons where both green and red fluorescence are present and the percentage of such neurons (per view) are counted (right, number of views in parentheses). **d** Subcellular distributions of YFP, GCaMP3 or NLS-GCaMP3-NLS overexpressed in cortical neurons. CFP fluorescence (NLS-CFP-NLS, upper) indicates the nucleus (blue) in cortical neurons. GFP images (lower row) of neurons expressing GCaMP3 (2 days after cDNA transfection) could be categorized into two major subgroups of interest: nuclear-excluded (N/C ratio < 0.6) and nuclear-filled (N/C ratio > 1.0), with the latter mimicked by neurons expressing NLS-GCaMP3-NLS. Such criteria (N/C ratio < 0.6 and >1) were applied throughout this study (unless indicated otherwise). **e** Based on the above criteria neurons of nuclear-filled group and nuclear-excluded group accounted for ~10% and ~50% of the total number of GCaMP3-expressing neurons, respectively (5 experiments). **f** Representative images tracing neurite morphology for cortical neurons from different subgroups. **g** Correlations between N/C ratio of GCaMP3 and neurite outgrowth. The grey line/area represents the total neurite length per neuron (control group). The eclipse enclosing most neurons contains two major areas representing the subgroups of nuclear-filled (cyan) and nuclear-excluded (pink). Nuclear GCaMP accumulation (N/C ratio) and neurite outgrowth (neurite length per neuron) are highly correlated (the correlation coefficient is −0.7). **h**, **i** Additional experiments and analyses for neurite morphology. Neurite outgrowth was quantified for the four subgroups by neurite length (**h**) and Sholl analysis (**i**), with the total number of cells in parentheses. Standard error of the mean (S.E.M.) and Student's *t*-test (two-tailed unpaired with criteria of significance: *$p < 0.05$; **$p < 0.01$, and $p < 0.001$) were calculated when applicable

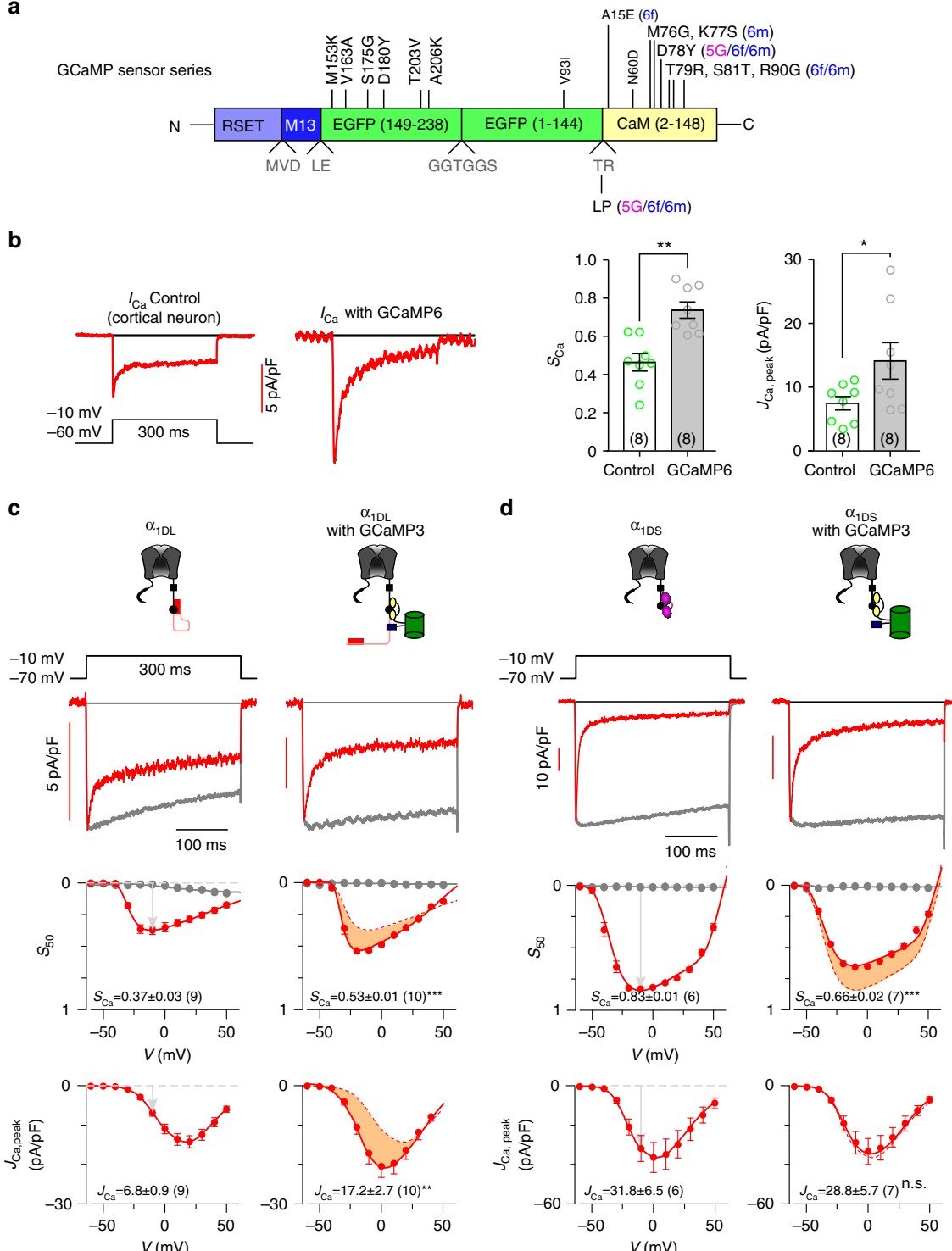

**Fig. 2** GCaMP interferes with Ca$_V$1.3 gating. **a** Schematic summary of GCaMP series. Upgrades of GCaMP (from GCaMP3 to GCaMP5 and GCaMP6) were achieved by mutations of the EGFP and CaM domains at the sites indicated by vertical letters (GCaMP3 vs. GCaMP), or by horizontal letters (GCaMP5G/6f/6m vs. GCaMP3). Details see Supplementary Fig. 1. **b** Effects on $I_{Ca}$ were examined for neurons infected with AAV-*Syn*-GCaMP6f. Representative traces of Ca$^{2+}$ current (left), $S_{Ca}$ and $J_{Ca}$ analyses (right) for native Ca$_V$1.3 in cortical neurons expressing GCaMP6. Neurons were treated with a cocktail recipe to isolate Ca$_V$1 current (mostly Ca$_V$1.3) and recorded at the membrane potential ($V$) of −10 mV. $S_{Ca}$ (quantified as $1-I_{Ca,50}/I_{Ca, peak}$, where $I_{Ca,50}$ and $I_{Ca, peak}$ represent the currents measured at 50 ms and the instantaneous peak, respectively) and $J_{Ca}$ (pA/pF, the current density of $I_{Ca, peak}$) serve as the indices for CDI and VGA respectively. **c** Effects of GCaMP3 on recombinant α$_{1DL}$. Representative Ca$^{2+}$ current traces were compared for $I_{Ca}$ recorded from HEK293 cells expressing long variant α$_{1DL}$ alone (left), or with GCaMP3 (right) at −10 mV. Ba$^{2+}$ currents (rescaled) and Ca$^{2+}$ currents ($I_{Ca}$) were shown as grey and red traces, respectively, with scale bars indicative of $I_{Ca}$ amplitudes. CDI ($S_{Ca}$) and VGA ($J_{Ca}$) profiles at different membrane potentials are compared between α$_{1DL}$ control and α$_{1DL}$ overexpressed with GCaMP3 (differences highlighted by orange areas). **d** Effects of GCaMP3 on recombinant α$_{1DS}$ alone (left), or with GCaMP3 (right), in a similar fashion to **c**. Standard error of the mean (S.E.M.) and Student's $t$-test (two-tailed unpaired with criteria of significance: *$p < 0.05$; **$p < 0.01$ and, ***$p < 0.001$) were calculated when applicable, and n.s. denotes "not significant"

expected, nuclear-filled neurons were subject to significant reduction in neurite length compared to control neurons (Fig. 1g, horizontal line in grey), indicative of neural damages. Surprisingly, neurite outgrowth appeared to be promoted in neurons where GCaMP3 was evidently excluded from the nucleus and the neurite length in average was significantly above the control level. Thus, a strong (negative) correlation was clearly formed between N/C ratio of GCaMP and the total neurite length per neuron (Fig. 1g). Nuclear accumulation and cell damages have been associated with the overexpression level of GCaMP, which was confirmed here by a similar correlation analysis but indexed with GCaMP fluorescence levels (Supplementary Fig. 3). These data suggest that GCaMP caused side-effects of at least two folds. First, as often noticed and reported in GCaMP applications, we here explicitly demonstrated that nuclear GCaMP, as one red alert sign along with the time or level of GCaMP expression, caused detrimental effects on neurons leading to neurite or cell loss. Second, discovered in this study, cytosolic GCaMP of less expression level still significantly altered neurite growth. In neurons without high-level GCaMP expression, such aberrant morphogenesis (indicative of altered signaling) could still be problematic to neurons even without clear sign of nuclear invasion. In quantitative comparison to the control group, the total neurite length was increased by cytosolic GCaMP3 and decreased by nuclear GCaMP3, the latter of which was further supported by reduction in neurite length of neurons expressing NLS-GCaMP3-NLS (Fig. 1h). Further comparisons among different groups of neurons by Sholl analysis consistently demonstrated that nuclear GCaMP reduced morphological complexity whereas cytosolic GCaMP promoted neurites to grow with higher complexity (Fig. 1i).

**GCaMP perturbs Ca$_V$1.3 gating.** Prior studies suspect that CaM contained in all versions of GCaMP (Fig. 2a and Supplementary Fig. 1) might be one factor responsible for various side-effects. However, it has been mostly attributed to trivial reasons, such as buffering effects of Ca$^{2+}$-binding CaM[4,16,26], one general concern common to any kind of Ca$^{2+}$ (binding) probes which fail to explain the multi-fold side-effects here observed. We accidentally acquired some evidence that GCaMP might perturb CaM in Ca$_V$1 gating[27], and thus Ca$_V$1-dependent signaling (e.g., excitation–transcription coupling)[28]. Ca$_V$1.3 is one major subtype of Ca$_V$1 channels expressed in neurons[29] and tightly coupled to transcriptional signaling in the nucleus[23,28]. Also, Ca$_V$1-mediated E–T coupling could control neurite outgrowth[30,31]. Importantly, Ca$_V$1.3 is subject to apoCaM tuning due to a competitive balance gauged by the strength of DCT and ambient concentrations of apoCaM[20,21]. Following these lines of evidence related to CaM and Ca$_V$1, we directly compared native $I_{Ca}$ of cortical neurons (mostly mediated by Ca$_V$1.3 in our protocol) between the control group and GCaMP6f group by whole-cell patch-clamp recording (Fig. 2b). Two major gating parameters were used to characterize Ca$^{2+}$ currents. Ca$^{2+}$-dependent inactivation (CDI) was quantified by strength/ratio factor $S_{Ca}$, and voltage-gated activation (VGA) was estimated by current density $J_{Ca}$ (details see Fig. 2b). $I_{Ca}$ recorded from GCaMP6f-overexpressing neurons was significantly different from the control, in that both CDI ($S_{Ca}$) and VGA ($J_{Ca}$) were augmented as shown by larger and "sharper" $I_{Ca}$ traces at −10 mV, raising serious concerns that GCaMP would perturb native Ca$_V$1 in neurons. Before pursuing further, we consolidated such distortions of Ca$_V$1 currents to dissect out the detailed mechanisms.

The aberrant Ca$_V$1.3 gating due to GCaMP roughly resembled the tuning profile of Ca$_V$1.3 by apoCaM competition against DCT[20,21]. We then checked GCaMP3 effects on recombinant

Ca$_V$1.3 channels in HEK293 cells. Ca$_V$1.3 long variant ($\alpha_{1DL}$), the major isoform of cortical Ca$_V$1, exhibited moderate CDI of intermediate $S_{Ca}$ which could be upregulated when free apoCaM is abundantly present. Notably, GCaMP3 acted on $\alpha_{1DL}$ channels in the fashion similar to WT apoCaM (Fig. 2c), that both CDI and VGA were significantly enhanced, consistent with the aforementioned effects on native $I_{Ca}$. On the other hand, does CaM as one motif of GCaMP really behave exactly the same as WT CaM? It is probably not the case, due to the following considerations. First, GCaMP of different versions all have certain mutations (very likely even more mutations in future versions) on CaM domain raising the possibility of defective CaM. Second, other domains of GCaMP may impair CaM functions. The CaM motif is surrounded with other motifs such as cpEGFP and M13 in GCaMP, which would structurally constrain the enclosed CaM domain from functioning as one 'real' CaM. In fact, as further analyzed with Ca$_V$1.3 and GCaMP variants, CaM motif in GCaMP has functional defects in both apoCaM and Ca$^{2+}$/CaM forms mainly due to cpEGFP and M13. With another isoform of Ca$_V$1.3 lacking DCT ($\alpha_{1DS}$)[32], we examined GCaMP3 effects on the gating (Fig. 2d). No change in VGA ($J_{Ca}$) was observed; however, the value of $S_{Ca}$ (CDI) exhibited a moderate but confirmative attenuation since normally CDI of $\alpha_{1DS}$ is very robust, maintaining at the constant level of ultrastrong CDI. Such alteration in CDI implied that Ca$^{2+}$/CaM functions of GCaMP were impaired. In principle apoCaM pre-association with the channel is required for subsequent Ca$^{2+}$/CaM-mediated Ca$_V$1 modulation. So how would GCaMP behave as apoCaM? GCaMP3 is able to win the competition against DCT or endogenous CaM to interact with the channel, shown by its effects on $\alpha_{1DL}$ (Fig. 2c) and $\alpha_{1DS}$ (Fig. 2d). However, further analysis demonstrated that GCaMP was only partially functioning as apoCaM because GCaMP with all the four Ca$^{2+}$-binding sites knocked out (GCaM$_{1234}$P) could not fully abolish CDI of $\alpha_{1DS}$ as expected from the Ca$^{2+}$-binding knockout CaM$_{1234}$ (Supplementary Fig. 4c), indicative of impaired apoCaM affinity to the channel. In support, for $\alpha_{1DS}$-DCT$_F$ channels with ultrastrong DCT effects, GCaMP barely exhibited any apoCaM tuning in comparison to WT CaM (Supplementary Fig. 4d). In all above experiments with defective apoCaM or Ca$^{2+}$/CaM from GCaMP, removal of M13 and cpEGFP eliminated the discrepant effects between WT CaM and CaM motif of GCaMP (Supplementary Fig. 4), also suggesting that the aforementioned potential effect of CaM mutations should be minimum, if there was any for GCaMP tested in this study.

In summary, GCaMP acts like CaM but with aberrant apoCaM and Ca$^{2+}$/CaM functionalities, suggested by GCaMP perturbations on Ca$_V$1.3 gating.

**GCaMP intervenes with excitation–transcription coupling.** The excitation–transcription coupling is highly specific to Ca$_V$1 in complex with CaM, which starts from Ca$_V$1 activities, followed by Ca$^{2+}$ binding to CaM proteins including the pre-associated CaM on the carboxyl terminus, Ca$^{2+}$/CaM signaling back to the channel (such as CDI), Ca$^{2+}$/CaM activation of CaMKII (CaM-dependent kinase II), CaM and CaMKII translocation to the nucleus, activation of nuclear CaM-dependent kinases and phosphorylation of CREB, eventually to gene transcription[23,28]. CaM plays essential roles in the whole E–T coupling, mediating Ca$^{2+}$ signaling to Ca$_V$1 and other important proteins in the cytosol and also the nucleus[21,24,33]. GCaMP perturbs the gating of Ca$_V$1/CaM complex (Fig. 2), and may also affect the downstream signals along the E–T pathway. We then examined the level of phosphorylated CREB (pCREB) in cultured cortical neurons expressing GCaMP3 of different distributions (Fig. 3a). pCREB

signals of each neuron were normalized by the mean intensity of pCREB fluorescence obtained from the control group. Normalized pCREB of GCaMP3-expressing neurons exhibited strong (negative) correlations with the N/C ratio (Fig. 3b) or expression level (Supplementary Fig. 5a) of GCaMP, resembling the high correlations previously obtained from neurite outgrowth (Fig. 1g and Supplementary Fig. 3). Grouped data from nuclear-excluded

(N/C ratio < 0.6) and nuclear-filled (N/C ratio > 1 or NLS-tagged) neurons revealed that pCREB signals under normal physiological conditions (control group) were significantly decreased by nuclear GCaMP3 but aberrantly enhanced by cytosolic GCaMP3 (Fig. 3c), while the total CREB levels remained the same for all neurons (Supplementary Fig. 5b). Such pCREB changes in dual directions were mostly attributed to GCaMP effects on native $I_{Ca}$

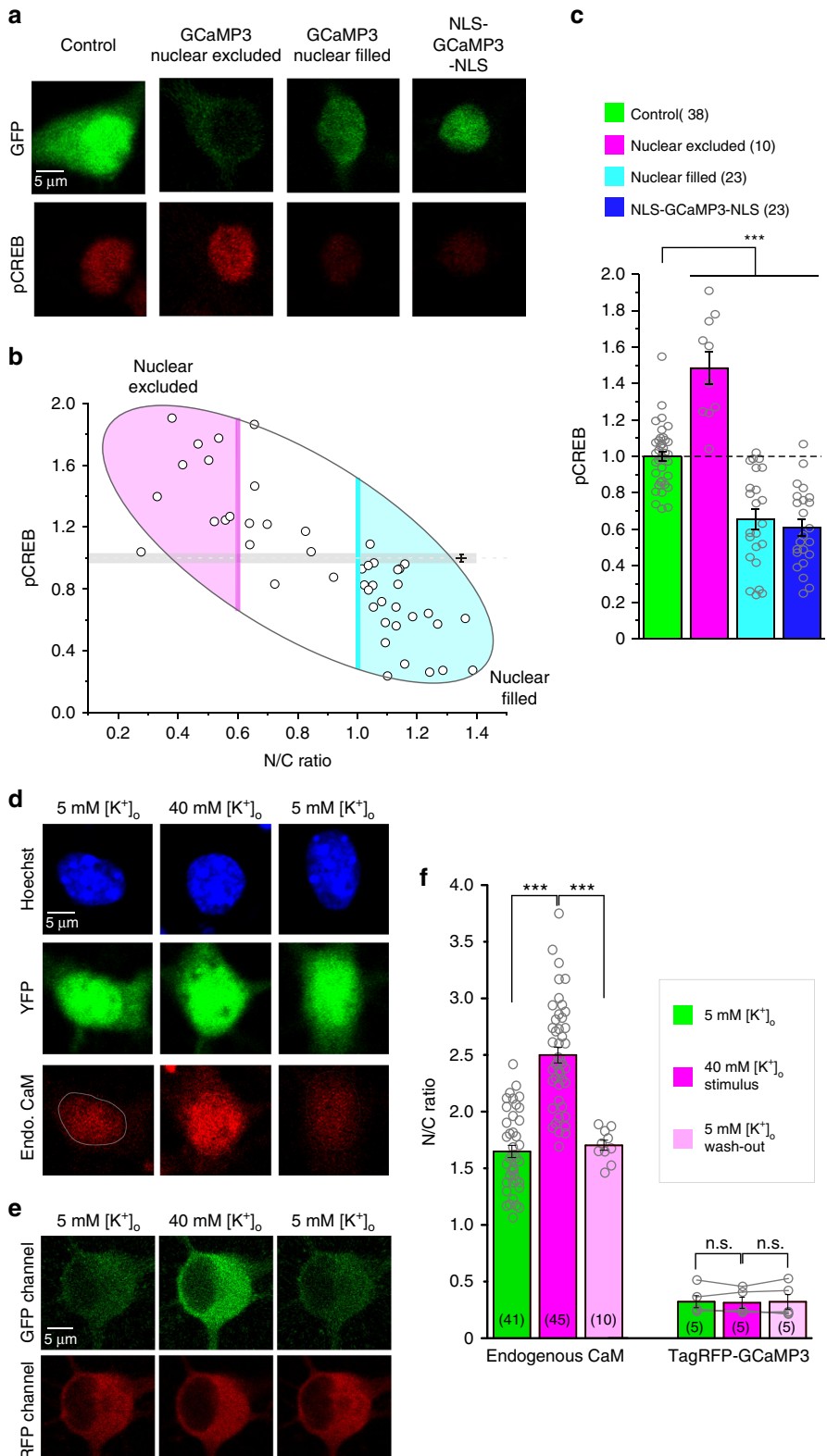

(Fig. 2b), providing the mechanistic linkage of nuclear/cytosolic GCaMP to aforementioned reduction/promotion of neurite branching and outgrowth (Fig. 1h). Particularly, GCaMP present in the nucleus could severely impair pCREB signaling by dominant negative effects of aberrant CaM-like activities, considering that CaM and CaM-interacting proteins plays important roles in nuclear signaling, many of which still await to uncover[24,34].

With enhanced membrane excitability by high $[K^+]_o$ stimulation, the potential perturbations of GCaMP on E–T coupling were examined. The majority of $Ca^{2+}$ entry into the neuron in response to 20–40 mM $[K^+]_o$ should be mediated by subthreshold $Ca_V1$ activities which subsequently drive transcriptional signaling including CaM translocation and CREB phosphorylation[23]. For control neurons, upon 40 mM $[K^+]_o$, endogenous CaM in the cytosol translocated to the nucleus, quantified by the increase of N/C ratio, which returned to the resting level after 5 mM $[K^+]_o$ washout (Fig. 3d, f). As the robust and critical event within E–T coupling in control neurons[28,33], cytonuclear translocation of CaM can serve as an acute form of index for transcription signaling since concurrently pCREB signals would be enhanced in response to high $[K^+]_o$ (Supplementary Fig. 6a–c). In contrast, GCaMP3 behaved rather differently from endogenous CaM, in that cytosolic GCaMP3 did not translocate into the nucleus upon high $[K^+]_o$, as indicated by red fluorescence from the fusion protein of TagRFP-GCaMP3 (Fig. 3e). The impaired translocation of GCaMP might arise from dysregulated CaM-dependent signal transduction to targeted proteins, similar to $Ca_V1.3$ perturbations. The corresponding pCREB signals, however, nearly lost the responses to high $[K^+]_o$ in neurons expressing GCaMP3, from either nuclear-excluded or nuclear-filled groups (Supplementary Fig. 6a–c). Acute translocation of endogenous CaM into the nucleus is also severely impaired in the neurons of nuclear-excluded group (Supplementary Fig. 6d). One plausible explanation is that cytosolic GCaMP could dominate the close vicinity to $Ca_V1$ channels, so that WT CaM is not (efficiently) recruited to initiate cytonuclear translocation. Notably, neurons of neither group could further respond to the stimulus in contrast to control neurons featured with activity-dependent CaM translocation (Fig. 3d, f) and CREB phosphorylation (Supplementary Fig. 6a–c), raising concerns on the potential impairment of neural plasticity and other subtle aspects of neural functions.

To this end, we discovered that GCaMP fundamentally altered E–T coupling regardless of its subcellular localization, mainly arising from its enclosed CaM mimicking the roles of WT CaM but with aberrant CaM functionalities, e.g., unexpected alternation of $Ca_V1$/CaM gating (cytosolic GCaMP), incapable of cytonuclear translocation, and perturbations on transcription signaling (nuclear GCaMP).

**Principles and implementations of GCaMP-X design.** Based on the perturbations of E–T coupling as the underlying mechanisms of GCaMP side-effects, we focused on the newly unveiled CaM-like roles of GCaMP to devise a new series of GCaMP-X sensors (Fig. 4a). The apoCaM binding onto $Ca_V1$ channels appeared to be the first and probably also the central problem. As the potential solution, after a trial-and-error process, we came up with the design of GCaMP-X by introducing an extra motif of apoCaM protection, originated from the IQ domain (dynamic CaM-binding domain) of neuromodulin (GAP-43)[35]. The protection motif was optimized to constitutively bind apoCaM with high affinity ($K_d \approx 2$ μM), but bind $Ca^{2+}$/CaM with much lower affinity ($K_d = 2 - 40$ μM) than M13 and other $Ca^{2+}$/CaM targets[36,37]. Such apoCaM binding motif (CBM) was fused onto the N-terminus of conventional GCaMP. At rest, CBM/apoCaM would form the complex thus eliminating the interferences with gating and signaling of $Ca_V1$/CaM complex. When $Ca^{2+}$ rises, high-affinity M13, but not CBM, would bind $Ca^{2+}$/CaM, relieving the concern that $Ca^{2+}$ sensing performance would be impaired by adding CBM. In addition to GCaMP-$X_O$ without any tag of localization signals, considering the particular importance of GCaMP distribution, we appended short tags onto the terminus of conventional GCaMP explicitly targeting different subcellular compartments (cytosol, membrane, and nucleus) as GCaMP-X variants of GCaMP-$X_C$, GCaMP-$X_M$, and GCaMP-$X_N$. To validate such design, the prototypes of new sensors were coexpressed with $Ca_V1.3$, in hope to alleviate the interference with its gating (Fig. 4b, c). With GCaMP3-$X_O$ or GCaMP3-$X_C$ being present, $α_{1DL}$ or $α_{1DS}$ channels behaved indistinguishably from the control (no coexpression of sensors). Their comparable values of $S_{Ca}$ (CDI) and $J_{Ca}$ (VGA) indices confirmed successful isolation of the sensors from $Ca_V1$. In addition to recombinant $Ca_V1.3$ in HEK293 cells, the design of GCaMP-X was also validated with native $I_{Ca}$ of cortical neurons infected with AAV viruses. The strong enhancement of CDI and VGA evidenced from conventional GCaMP (AAV-*Syn*-GCaMP6f) was diminished when GCaMP-X (AAV-*Syn*-GCaMP6m-$X_C$) was applied to neurons instead (Fig. 4d).

Neurons expressing GCaMP6m produced significantly larger fluorescence signals (index with $\Delta F/F_0$, fold of change in fluorescence) and lasted longer period of time upon high $[K^+]_o$ stimulation, suggesting more $Ca^{2+}$ influx in comparison with GCaMP6m-$X_C$ (Fig. 4e), if similar sensing performance could be assumed for both GCaMP6m and GCaMP6m-$X_C$ (later proved). In addition, ratiometric $Ca^{2+}$ fluorescence imaging with Fura-2 (5 μM) was performed, where $F_{340}/F_{380}$ ratio ($F_{340}$ and $F_{380}$, the fluorescence intensities of Fura-2 at excitation wavelength of 340 nm or 380 nm) was quantified as the index of (relative) $Ca^{2+}$ concentration. Upon 40 mM $[K^+]_o$ applied to neurons,

**Fig. 3** GCaMP perturbs $Ca_V1$-dependent E–T coupling. **a** GCaMP3 effects on CREB signaling. Cortical neurons (DIV 5) were transfected with YFP, GCaMP3 or NLS-GCaMP3-NLS for 2 days, then stained with pCREB antibodies under basal conditions. Green fluorescence represents the distributions of protein expression (upper) and red staining represents the levels of pCREB signals (lower), for all the four groups of neurons: YFP control, GCaMP3 nuclear excluded, GCaMP3 nuclear filled, and NLS-GCaMP3-NLS. **b** High correlation between subcellular localization of GCaMP3 (N/C ratio) and pCREB signals in cortical neurons expressing GCaMP3 (correlation coefficient = −0.8). Shades of areas indicate the neurons in the nuclear-excluded group (pink) and the nuclear-filled group (cyan). **c** Statistical summary of basal pCREB intensities. All neurons were normalized by the average value of pCREB fluorescence intensities calculated from the control group. **d**–**f** Cytonuclear translocations of CaM and GCaMP3 by confocal imaging. Endogenous CaM of cortical neurons translocated into nucleus under 5 min of 40 mM $[K^+]_o$ stimulation, and recovered to the resting state when washed back by 2 h of 5 mM $[K^+]_o$ (**d**). Representative images of fixed neurons for Hoechst (blue, indicating nucleus), YFP (indicating soma) and endogenous CaM (red, staining of CaM antibodies) at three different time points are shown in order (before 40 mM $[K^+]_o$, at the end of 5 min of 40 mM $[K^+]_o$ stimulation, after washed back to 5 mM $[K^+]_o$ for 2 h). 1 μM TTX was added throughout the experimental protocol in order to examine subthreshold activities of $Ca_V1$ channels. For comparison, neurons expressing TagRFP-GCaMP3 were similarly excited and monitored, during which localizations of GCaMP3 were confirmed by fluorescent TagRFP (**e**). RFP fluorescence at three representative time points (resting, stimulation, and wash-out) was analyzed with the index of N/C ratio and compared (paired Student's *t*-test) for GCaMP3 localization, in addition to endogenous CaM (**f**). Standard error of the mean (S.E.M.) and Student's *t*-test (two-tailed unpaired with criteria of significance: *$p < 0.05$; **$p < 0.01$, and $p < 0.001$) were calculated when applicable, and n.s. denotes "not significant"

conventional GCaMP resulted into aberrantly higher $Ca^{2+}$ signals than GCaMP6m-$X_C$ (Fig. 4f). We assume that both GCaMP and GCaMP-X of similar expression levels (tens of μM) would also produce "buffering effects" about equally in average. So for the neurons loaded with 5 μM Fura-2, adding GCaMP or GCaMP-X would further attenuate the signals to the same extent. And importantly, in the context of such fair comparison between GCaMP and GCaMP-X, the larger $Ca^{2+}$ signal should be attributed to abnormal GCaMP enhancement of $Ca^{2+}$ currents mediated by $Ca_V1$ channels. Consistent with electrophysiological recordings, $Ca^{2+}$ fluorescence imaging with GCaMP itself or organic dye Fura-2 demonstrated that

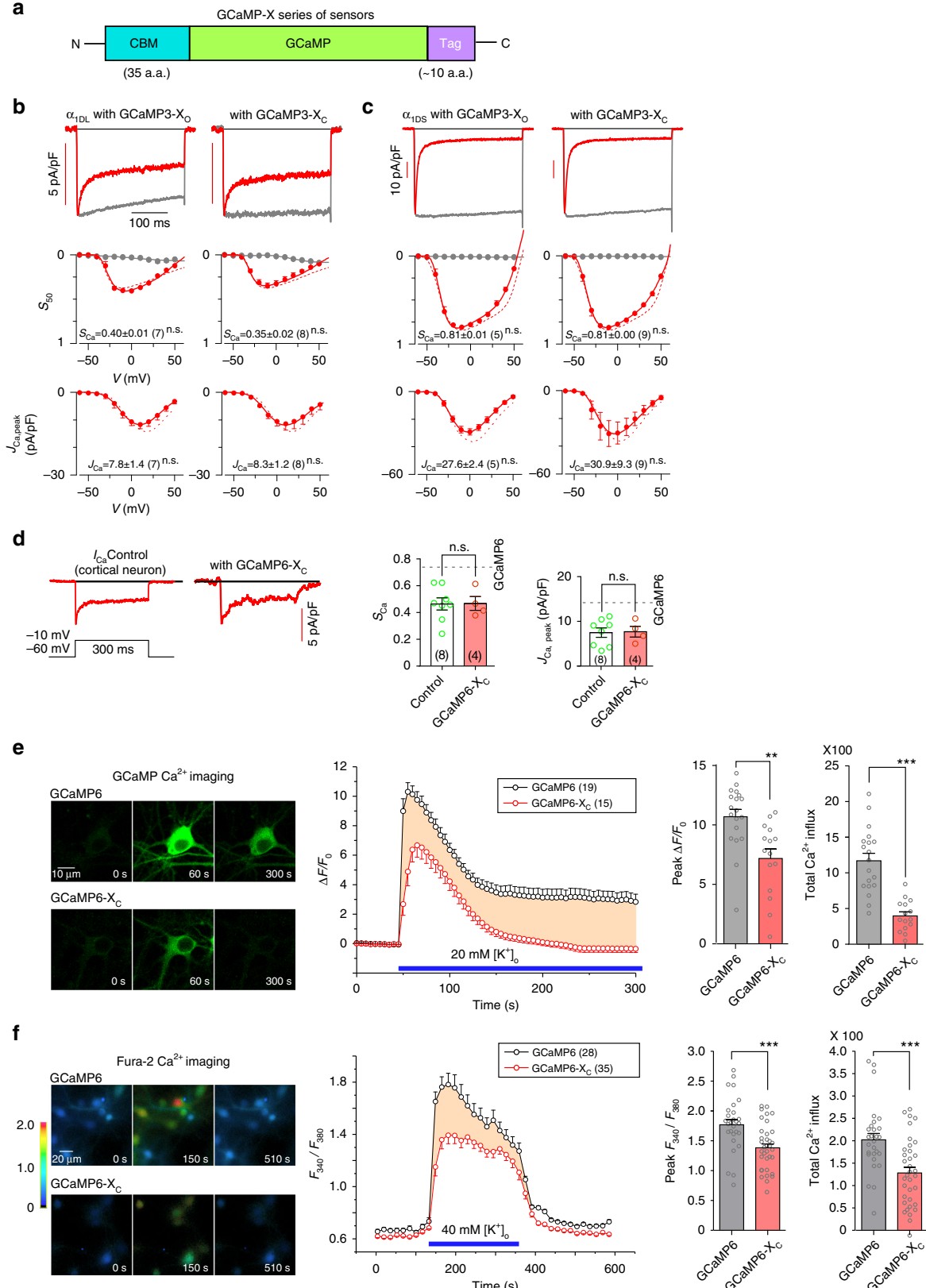

$Ca^{2+}$ dynamics of neurons would be perturbed by GCaMP acting on $Ca_V1$ channels, as further confirmed by isradipine blockage (Supplementary Fig. 7). These nontrivial alterations in transmembrane and cellular $Ca^{2+}$ called up attentions to carefully analyze/revisit the data and conclusions acquired by conventional GCaMP, due to the importance of $Ca^{2+}$ itself and also its critical roles in multiple physiological processes, in particular $Ca_V1$-dependent E–T coupling. Membrane-targeted version of GCaMP could be particularly useful for certain applications such as monitoring astrocyte calcium[38], and here we implemented GCaMP3-$X_M$ by incorporating the membrane-tethering Lck tag at the N-terminus of GCaMP3-X. Similar to GCaMP3-$X_C$ vs. GCaMP3, GCaMP3-$X_M$ in comparison to GCaMP3$_M$ also corrected the aberrant enhancement of $\alpha_{1DL}$ gating and overall $Ca^{2+}$ influx, demonstrated by patch-clamp recordings and Fura-2 fluorescence imaging (Supplementary Fig. 8).

Thus, GCaMP-X eliminated the problems of $Ca^{2+}$ distortion in the first place, which promised to alleviate aforementioned GCaMP perturbations on $Ca^{2+}$ signaling.

**GCaMP-X no longer perturbs $Ca_V1$ signaling in neurons.** Subsequent to validation of GCaMP-X in the aspects of $Ca_V1$ gating, we examined the side-effects of conventional GCaMP on other processes of E–T coupling in cortical neurons (Figs. 1 and 3). Firstly, distribution analysis indexed with N/C ratio for GCaMP3, GCaMP3-$X_O$, and GCaMP3-$X_C$ indicated that GCaMP3-$X_O$ or GCaMP3-$X_C$ no longer accumulated in the nucleus in contrast to GCaMP3 (Fig. 5a). Intriguingly, although fusion of a NES (nuclear export signal) tag did ensure the cytosolic localization, GCaMP-$X_O$ itself already very much relieved the abnormal nuclear accumulation. The reason responsible for abnormal GCaMP nuclear accumulation is not very clear yet. Basic properties of the protein itself may play a role, such as the size and shape. Or alternatively, such abnormality could be due to some unexpected interference with normal $Ca_V1$-dependent cytonuclear translocation processes. GCaMP-X protected apoCaM and subsequent $Ca^{2+}$/CaM from interactions with endogenous proteins (e.g., $Ca_V1$ channels) in cells, as the potential mechanism to account for its cytosolic retention even without NES tagging (GCaMP-$X_O$). Nevertheless, explicit tags should still be used if possible to explicitly control GCaMP-X distributions in different cellular conditions or contexts. Consistent with the profiles of N/C ratios, no more perturbation on pCREB signals (Fig. 5b) and neurite outgrowth (Fig. 5c) could be observed when GCaMP-$X_O$ or GCaMP-$X_C$ were overexpressed in cortical neurons.

Furthermore, we examined the neurons infected with AAV-$Syn$-GCaMP6m-$X_C$ vs. conventional GCaMP. Different from AAV-$Syn$-GCaMP6f (Fig. 1b), long-term infection of AAV-$Syn$-GCaMP6m-$X_C$ no longer caused the time-dependent nuclear accumulation in neurons; instead, rather stable cytosolic distributions were maintained: N/C ratio was $0.36 \pm 0.01$ ($n = 69$) even after three weeks (Fig. 5d). And for apoptosis associated with GCaMP, neurons of 10 days after virus infection of AAV-$Syn$-GCaMP6m-$X_C$ exhibited much weaker apoptotic signals of fluorescent Annexin V (<20% neurons), almost back to the level of control neurons (Fig. 5e), in contrast to the pronounced damages by GCaMP6f (>60%, Fig. 1c). To this end, all the known side-effects related to $Ca_V1$-dependent transcriptional signaling were all eliminated or strongly attenuated by our newly designed GCaMP-X.

In addition, we systematically examined other versions of GCaMP (e.g., GCaMP5G and GCaMP6m) also other representative CaM-based GECIs including D3cpv, CaMPARI and inverse pericam; and confirmed that all of them exhibited similar problems such as perturbations of $Ca_V1.3$ gating (both $\alpha_{1DS}$ and $\alpha_{1DL}$), abnormal nuclear accumulations, or alterations of neuronal morphology (Supplementary Fig. 9 and Supplementary Fig. 10). Improved sensors such as GCaMP5-$X_C$ exhibited no or little side-effects on $Ca_V1$ gating and signaling in neurons, which not only further validated our design principle, but also strongly implied its generalizability to all other CaM-based GECIs.

As summarized in the scheme (Fig. 6a), GCaMP defects and mechanisms are clarified by this work, as the basis for the new design of GCaMP-X. In neurons, GCaMP sensors behave as CaM-like proteins. In consequence, $Ca_V1$ gating is perturbed through apoCaM and subsequent $Ca^{2+}$/CaM modulation, causing unexpected increase of $Ca^{2+}$ influx and distorted channel kinetics, accumulated nuclear GCaMP, impaired cytonuclear CaM translocation, aberrant gene transcription, and dysregulated growth of neurites. The general health of neurons is impaired and overall $Ca^{2+}$ dynamics is altered, both of which would affect sensor readouts. In contrast, the GCaMP-X sensors containing extra protective motif of apoCaM-binding CBM successfully avoid all the above problematic effects on $Ca_V1$-dependent E–T coupling.

**Sensor performance of GCaMP-$X_C$ is comparable to GCaMP.** GCaMP-X does not introduce any de novo mutations on the part of GCaMP and CBM would not interfere with $Ca^{2+}$/CaM binding of M13 within GCaMP, thus unlikely to affect its performance as $Ca^{2+}$ sensors. Due to GCaMP perturbations on E–T coupling in excitable cells (which cause $Ca^{2+}$ distortion in the first place), instead of native neurons, HEK293 cells in response to 10 μM

**Fig. 4** Design principles and basic validations of new GCaMP-X sensors. **a** Design principles of GCaMP-X. CaM binding motif (CBM) was fused onto N-terminus of GCaMP to tightly bind apoCaM at rest but with relatively low affinity (e.g., much lower than the M13 motif) to $Ca^{2+}$/CaM, as the central strategy for GCaMP-X design. Also, additional tags of NES (nuclear export signal), NLS (nuclear localization signal) and MTS (membrane targeting signal) motifs could be appended onto C-terminus (or N-terminus), to control specific subcellular localization of GCaMP. Accordingly, a series of GCaMP-X sensors were developed, in addition to GCaMP-$X_O$ without any tag, including GCaMP-$X_C$, GCaMP-$X_N$, and GCaMP-$X_M$, tagged with NES, NLS, and MTS respectively. **b**, **c** Design validation with GCaMP3 and recombinant $Ca_V1.3$ channels. Neither CDI nor VGA of $\alpha_{1DL}$ channels (**b**) and $\alpha_{1DS}$ channels (**c**) were altered by GCaMP3-$X_O$ or GCaMP3-$X_C$ overexpressed in HEK293 cells, as demonstrated by $I_{Ca}$ exemplars (top) and voltage-dependent CDI (middle) and VGA (bottom) profiles. **d** Validation of GCaMP6m-$X_C$ with neuronal $Ca_V1.3$ channels. Effects of GCaMP6-$X_C$ on $\alpha_{1DL}$ gating were examined in cortical neurons infected with AAV-$Syn$-GCaMP6m-$X_C$, indexed with $S_{Ca}$ and $J_{Ca}$. Dotted lines represent GCaMP6f profiles (Fig. 2b). **e**, **f** GCaMP-X eliminated perturbations on subthreshold $Ca^{2+}$ dynamics. For cortical neurons infected with AAV-$Syn$-GCaMP6m or AAV-$Syn$-GCaMP6m-$X_C$, confocal fluorescent images depict $Ca^{2+}$ dynamics in response to extracellular stimuli of 20 mM $[K^+]_o$, indexed by normalized fluorescence changes ($\Delta F/F_0$). Exemplar images (left), averaged curves of $\Delta F/F_0$ (middle) and statistical summary (right) are shown for GCaMP6m and GCaMP6m-$X_C$. Particularly, peak $\Delta F/F_0$ and total calcium influx (rightmost, area integral within 300 s, in the units of $\Delta F/F_0 \cdot$s) were evaluated and compared (**e**). To confirm, similar experiments and analyses were performed with Fura-2 ratiometric $Ca^{2+}$ imaging, indexed with fluorescence emission ratio $F_{340}/F_{380}$ achieved from 340 nm and 380 nm excitation (**f**). Standard error of the mean (S.E.M.) and Student's $t$-test (two-tailed unpaired with criteria of significance: $*p < 0.05$; $**p < 0.01$, and $***p < 0.001$) were calculated when applicable, and n.s. denotes "not significant"

acetylcholine (Ach)[5] were used as the cellular context to fairly compare basic sensor characteristics such as $Ca^{2+}$ sensitivities, which turned out to be indistinguishable between GCaMP and GCaMP-$X_C$ (Supplementary Fig. 11a). To closely examine the kinetics, the approach we employed was to induce $Ca^{2+}$ dynamics mimicking that of one single action potential (AP), by fast break-in with brief ZAP stimulus, aided with strong $Ca^{2+}$ chelators of

10 mM BAPTA in patch-recording pipettes. This way, a $Ca^{2+}$ transient was created with fast onset and offset. GCaMP6m and GCaMP6m-$X_C$ resulted into indistinguishable characteristics of peak $\Delta F/F_0$, SNR, rise time $t_r$ and decay time $t_d$ (Fig. 6b). Their $t_r$ values were about the same (~0.1 s), further confirmed by an alternative approach to induce faster ($t_r < 0.1$ s) $Ca^{2+}$ influx via voltage-gated $Ca_V2.2$ channels (Supplementary Fig. 12), which

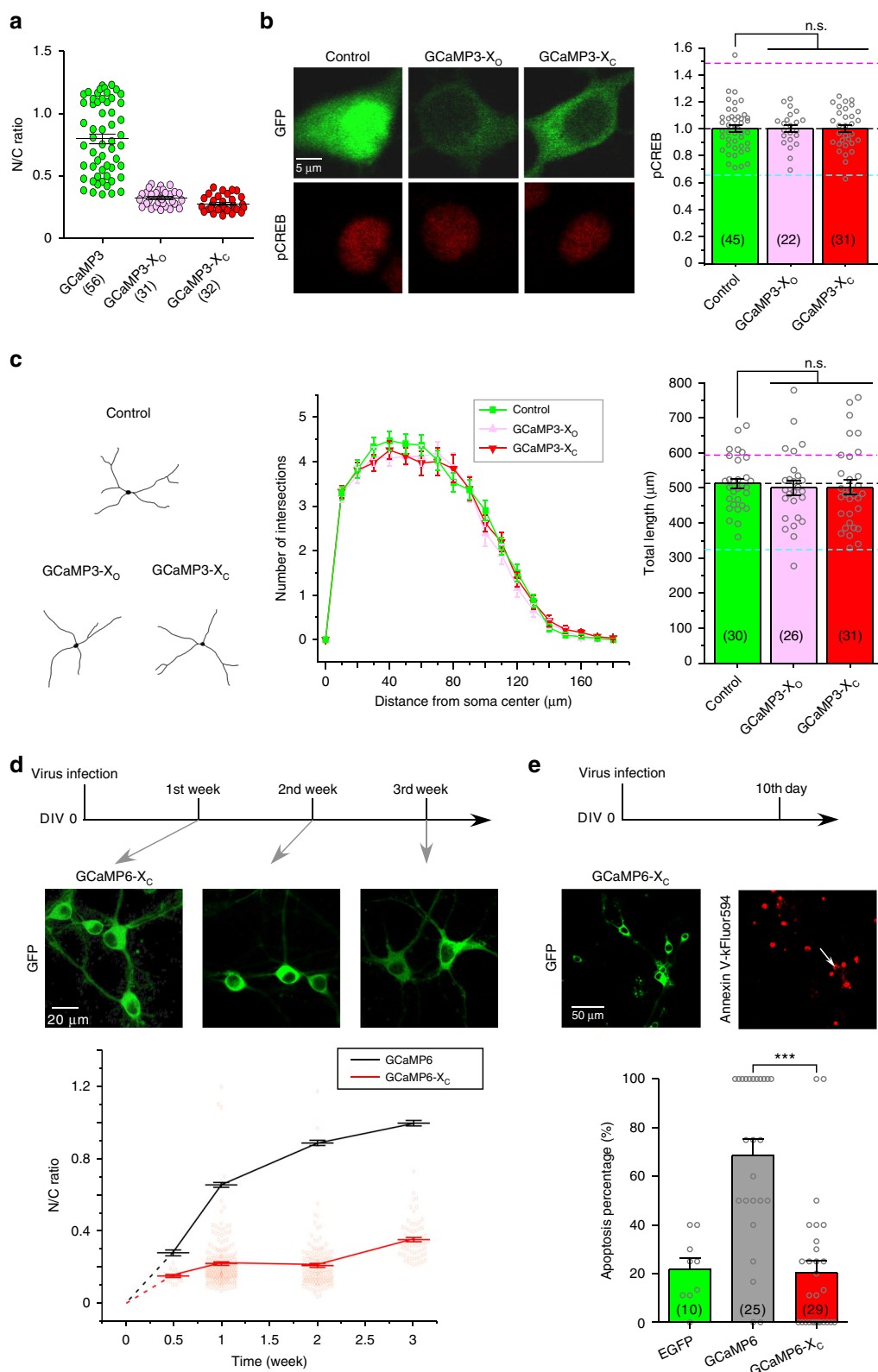

are similar to the measurements from single AP-driven $Ca^{2+}$ in neurons[7]. Their comparable sensor performances suggest that $Ca^{2+}$-dependent conformational changes within GCaMP-X should have no major difference from conventional GCaMP.

In addition, for outer hair cells expressing GCaMP6m or GCaMP6m-$X_C$ (by electroporation), $Ca^{2+}$ fluorescence indicative of mechanosensitive currents further confirmed that GCaMP-X design would not perturb $Ca^{2+}$ sensing mechanisms within GCaMP. Upon mechanical stimuli of fluid jet with different durations (100 ms, 300 ms or 500 ms)[39] (Fig. 6c), freshly-prepared tissue and cells (DIV 1) exhibited similar stimulus-dependent fluorescence signals ($\Delta F/F_0$) for both sensors (Fig. 6d). However, for a loner time of tissue culture (DIV 3), during which the side-effects of GCaMP6m (e.g., by interfering with $Ca_V1.3$ in hair cells) would emerge, manifested as weaker $Ca^{2+}$ responses as compared with cells expressing GCaMP6m-$X_C$, supporting that GCaMP6m caused the attenuation of $Ca^{2+}$ signals while GCaMP6m-$X_C$ avoided the problem (Fig. 6e).

Therefore, GCaMP-X inherits the excellent properties of $Ca^{2+}$ sensing evolved from generations of GCaMP, while avoiding $Ca^{2+}$ distortion and other problems. Documented problems of GCaMP in $Ca^{2+}$ sensing, such as weak or diminished readout of $Ca^{2+}$ signal, are likely due to E–T perturbations and $Ca_V1$ dysregulations in excitable cells.

**GCaMP-$X_N$ as a novel type of nuclear $Ca^{2+}$ sensors.** Nuclear $Ca^{2+}$ plays vital roles in the nervous system[34], the heart[40], and immune cells[41]. In neurons, nuclear $Ca^{2+}$ could act as the controlling factor over transcription signaling and gene expression. Organic fluorescent dyes such as Fura-2 may help explore $Ca^{2+}$ dynamics in the nuclear[42]. However, for GECIs, due to aforementioned side-effects and mechanisms, serious concerns have been raised regarding whether CaM-based GECIs including GCaMP can serve as safe and trustworthy probes when present in nuclei. These effects sometimes were apparent since nuclear-filled GCaMP seemed to end up with distorted and attenuated fluorescence, indicative of dysregulated $Ca^{2+}$ dynamics[5] due to cell damages. In some other cases, such side-effects of nuclear GCaMP might be less noticeable, e.g., effects of NLS-GCaMP-NLS earlier before any apparent damage to the cell, because the interference with E–T signaling should be always present in principle and also takes time to manifest (Figs. 1h, 3c and Supplementary Fig. 6c).

Greatly encouraged by GCaMP-X prototypes which successfully eliminate deleterious side-effects, we proceeded to design a new type of GCaMP-$X_N$ targeting genuine and intact nuclear $Ca^{2+}$ without aforementioned perturbations (Fig. 7a). GCaMP-$X_N$ is based on GCaMP-X but with NLS (nuclear localization signal) motif fused onto the C-terminus. We first validated GCaMP-$X_N$ by examining and comparing pCREB signals and neurite outgrowth in cortical neurons transfected with NLS-GCaMP3-NLS vs. GCaMP3-$X_N$. Neurons expressing GCaMP3-$X_N$ exhibited similar levels of pCREB and neurite outgrowth as the control neurons, whereas NLS-GCaMP3-NLS exhibited strong pCREB inhibition and caused neurite damages (Fig. 7b, c), demonstrating the advantages of GCaMP-$X_N$ over conventional GCaMP as sensors of nuclear $Ca^{2+}$. In cortical neurons stimulated by 40 mM $[K^+]_o$, GCaMP3-$X_N$ with confocal fluorescence imaging resulted into more pronounced (larger peak of $\Delta F/F_0$) and faster (8-fold rising speed in $\Delta F/F_0 \cdot s^{-1}$) $Ca^{2+}$ dynamics than NLS-GCaMP3-NLS (Fig. 7d). In line with prior reports, high correlations existed between cytosolic (by GCaMP-$X_C$) and nuclear $Ca^{2+}$ signals (by either GCaMP-$X_N$ or NLS-GCaMP-NLS) (Fig. 7e, f). For neurons expressing GCaMP-$X_N$, GCaMP-$X_C$ or GCaMP (nuclear-filled or nuclear-excluded), the comparison among their $Ca^{2+}$ responses suggest that GCaMP-$X_C$ and GCaMP-$X_N$ should provide most reliable measurements, while GCaMP distorted $Ca^{2+}$ signals in dual directions, corresponding to attenuated or enhanced $Ca_V1$ and E–T coupling in nuclear-filled or nuclear-excluded neurons respectively (Supplementary Fig. 13). All these data suggest that the discrepancy between the new GCaMP-$X_N$ and conventional nucleus-localized GCaMP should be mainly due to systematic dysregulations of $Ca^{2+}$ dynamics and signaling rather than intrinsic sensing capabilities of the probes. We provided further evidence with HEK293 cells which do not have the neural mechanisms of $Ca_V1$-dependent E–T coupling. In HEK293 cells responding to Ach stimulation, similar cytosolic $Ca^{2+}$ signals were measured by GCaMP-$X_C$ and GCaMP; and GCaMP-$X_N$ and NLS-GCaMP-NLS resulted into comparable readouts of nuclear $Ca^{2+}$ (Supplementary Fig. 11b).

## Discussion

In this work, we conducted in-depth analyses on the mechanisms of the "side-effects" existing to widely-applied GCaMP sensors in $Ca^{2+}$ fluorescence imaging. Data unveiled that GCaMP interferes with $Ca_V1$/CaM-mediated excitation–transcription coupling in neurons. With these novel insights, we then developed new GCaMP-X sensors immune to side-effects for faithful monitoring of $Ca^{2+}$ dynamics in cells including specific subcellular compartments, e.g., cytosol (GCaMP-$X_C$) or nucleus (GCaMP-$X_N$) or plasma membrane (GCaMP-$X_M$). The key is to supply GCaMP with an extra motif of high-affinity apoCaM-binding, which eliminates GCaMP perturbations on apoCaM and its subsequent $Ca^{2+}$/CaM signaling endogenous to the native cell or particularly its $Ca_V1$ channels. This way, GCaMP-X sensors resolved all the problems of GCaMP, while still inheriting its excellent sensing capabilities in monitoring $Ca^{2+}$ dynamics.

**Fig. 5** Additional validations of GCaMP-X. **a** Nuclear accumulation was substantially relieved for cortical neurons transfected with GCaMP-X. N/C fluorescence ratios indicative of GCaMP distributions in cortical neurons (DIV 5) were quantified 2 days after being transfected with GCaMP3, GCaMP3-$X_O$ or GCaMP3-$X_C$. **b** GCaMP-X sensors no longer perturbed pCREB signals. For cortical neurons transfected with YFP, GCaMP3-$X_O$ or GCaMP3-$X_C$, fluorescence of pCREB immunostaining (left) was normalized and compared (right). Dotted lines represent the mean values from nuclear-excluded GCaMP3 (pink) or nuclear-filled GCaMP3 (cyan) neurons, respectively (adopted from Fig. 3c). **c** Dysregulations of neurite outgrowth were diminished with GCaMP-X. Based on neurite tracing images (left), Sholl analysis (middle) and measurement of neurite length (right) were performed for cortical neurons transfected with YFP, GCaMP3-$X_O$ or GCaMP3-$X_C$. Similar to **b**, dotted lines depict mean values of neurite length for nuclear-excluded GCaMP3 (pink) or nuclear-filled GCaMP3 (cyan) neurons (adopted from Fig. 1h). **d** Problematic nuclear accumulations were strongly attenuated for long-term viral expression of GCaMP6m-$X_C$. Fluorescence indicative of sensor localization was mostly restricted to the cytosol of neurons infected with AAV-Syn-GCaMP6m-$X_C$ (GCaMP6-$X_C$), as shown by representative images at different time points (upper panel). The temporal trend of N/C ratio for GCaMP6-$X_C$ (red) is distinct from that for GCaMP6f (black, GCaMP6 data adopted from Fig. 1b). **e** Apoptotic damages arising from GCaMP were strongly attenuated in GCaMP-$X_C$ expressing neurons. Fluorescence signals of apoptosis (red) were detected by Annexin V assay, to correlate with the expression of sensors (green). AAV-Syn-GCaMP6m-$X_C$ (GCaMP6-$X_C$), EGFP and GCaMP6f (data adopted from Fig. 1c) were compared in a similar fashion to Fig. 1c. Standard error of the mean (S.E.M.) and Student's $t$-test (two-tailed unpaired with criteria of significance: $*p < 0.05$; $**p < 0.01$, and $***p < 0.001$) were calculated when applicable, and n.s. denotes "not significant"

CaM has been encoded into various CaM-based GECIs, with GCaMP as the representative. Vast efforts were put into updating GCaMP generation by generation through screening thousands of structure-based or random mutagenesis to improve baseline fluorescence, photostability, dynamic range, and $Ca^{2+}$ affinity[5–7,43]. However, until this work the major drawback of GCaMP has not yet been specifically addressed and resolved, such as the well-known nuclear accumulation and related cell damages. We went beyond simply attributing these side-effects to over-buffering which supposedly could happen to almost any probes of excessive amount[15,16], unveiling that the true mechanisms underlying GCaMP effects are due to its CaM motif, which interferes with $Ca_V1$-dependent E–T coupling thus damaging neurons (Figs. 1–3 and Fig. 6a).

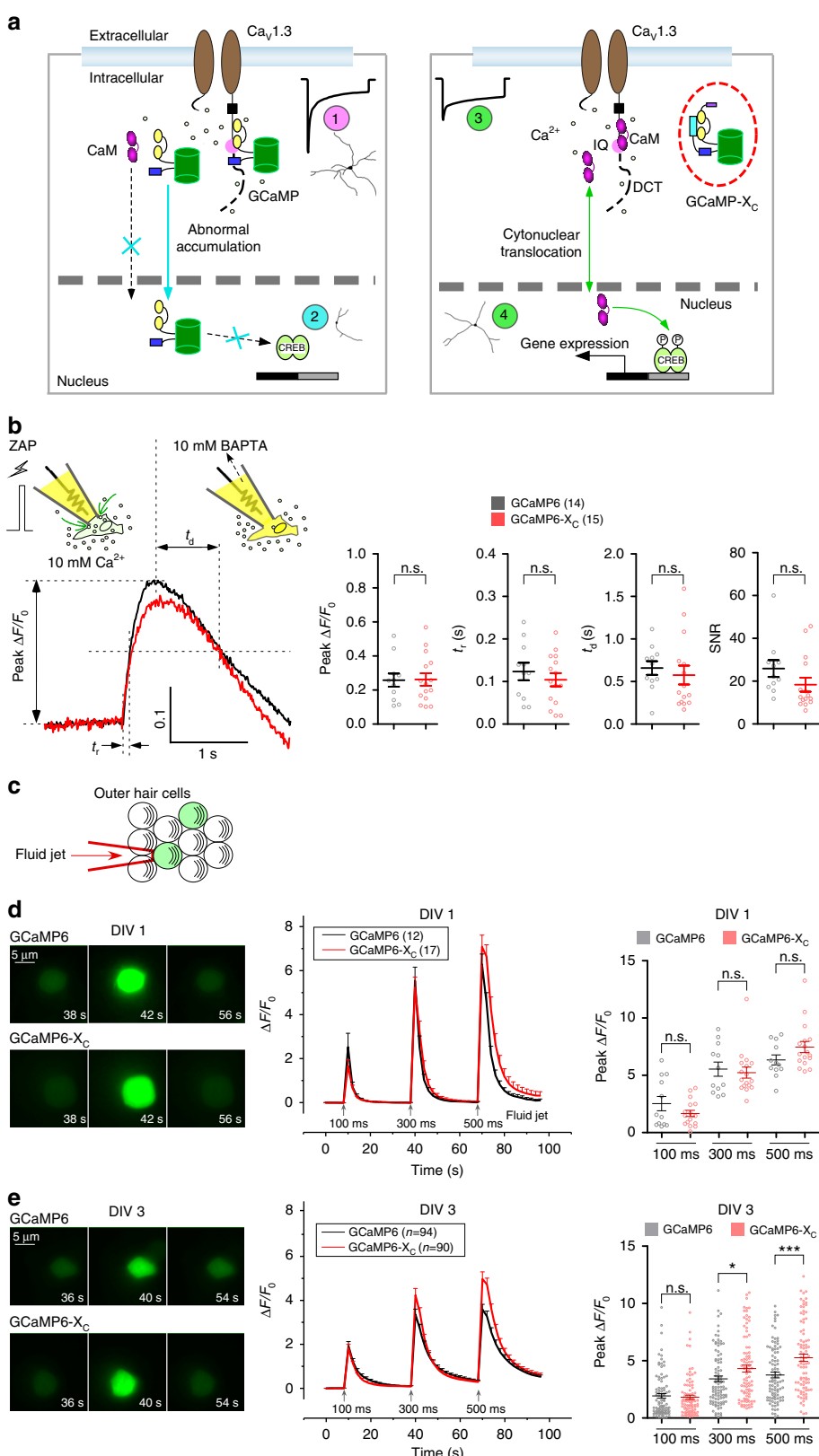

Meanwhile, side-effects of another kind, although less obvious, could significantly alter intrinsic $Ca^{2+}$ dynamics and membrane excitability (Fig. 4, Fig. 6e and Fig. 7f) as well as cellular signaling (Fig. 3), which could well explain similar observations from neurons and myocytes[4,9,13,14,44]. Nevertheless, perturbations of this kind have not been widely reported thus far, probably due to the following reasons. First, effects of neural damage and cell death are more apparent and problematic than other less discernible phenotypes such as $Ca_V1$ upregulations. Second, the basal fluorescence of $Ca^{2+}$-free GCaMP is far less bright than GFP or $Ca^{2+}$/GCaMP. Thus, relying on the basal GCaMP fluorescence, the neurons categorized as "low expression" may still have substantial levels of GCaMP generating the phenotypes of $Ca_V1$ upregulation; meanwhile, GCaMP below this threshold level would be overlooked in the experiments due to "no expression".

Our discovery of $Ca_V1$/GCaMP interplays essentially provides the mechanistic linkage to the well-known (damages of cell health) and less-noticeable (alterations of cell functions) drawbacks of CaM-based GECIs. Based on the phenotypical correlation between neural damages and GCaMP levels, existing efforts tried to gauge GCaMP expression with different strategies, e.g., to select relatively weak promotors trying to alleviate the side-effects. Abnormalities in E–T coupling in GCaMP-expressing neurons should happen earlier than the clear sign of nuclear GCaMP accumulation and neural damages (Supplementary Fig. 6d). Moreover, GCaMP-X without any localization signal, even with high expression levels, no longer accumulated in the nucleus to perturb pCREB signals (Fig. 5a, b), suggesting apoCaM protection is an effective way to alleviate nuclear filling. Considering the direct effects of cytosolic and nuclear GCaMP, it is likely that multiple CaM/GCaMP-dependent processes are involved in aberrant nuclear invasion and subsequent cell damages. When these processes were not present or less dominant, e.g., in HEK cells lacking native E–T coupling, no difference between GCaMP and GCaMP-X was found (Supplementary Fig. 11), distinct from neurons (Fig. 7).

Some current lines of transgenic mice using weaker promoters suffer from low expression levels, limiting their usage due to poor SNR[16]. GCaMP6 mice should be much improved in that the sensing performance is excellent while exhibiting no sign of nucleus filling as claimed[45]. However, GCaMP present in the cytosol is still problematic by acting on $Ca_V1$. In comparison with viral infection, nucleus-filling neurons from GCaMP6 transgenic mice are much less, but still discernible[46]; and nuclear GCaMP of even lesser amount is still able to intervene gene transcription in principle. Evidently, GCaMP affected neural excitability in GCaMP5G or GCaMP6 transgenic mice[13,14]. Although no difference in LTP was detected with or without GCaMP3 in CA1 hippocampal neurons[47], further thorough analyses of synaptic plasticity are necessary because subtle but confirmative effects on E–T coupling by GCaMP could vary, depending on particular brain regions, cell types, expression levels and experimental settings.

Non-CaM based sensors, such as the troponin-based TN-XL series[48], are unlikely to suffer from the side-effects associated with GCaMP or aberrant CaM. Meanwhile, many CaM-based GECIs have been developed and still currently used, such as GCaMP6f/m/s and GCaMP variants of other colors (e.g., RCaMP), cameleons (FRET-based GECI), D3cpv and Camgaroos (improved versions of cameleons), CaMPARI (a newly reported GECI of Calcium Modulated Photoactivatable Ratiometric Integrator), inverse pericam (fluorescence inversely correlating with $Ca^{2+}$) and so on[6,7,49–54]. As proved with some of these GECIs, E–T coupling would be similarly perturbed by the CaM domain of the above sensors through apoCaM and/or $Ca^{2+}$/CaM effects, leading to functional and/or morphological defects in cells. It is implied that the design principles should be even more broadly applicable beyond GCaMP-X, serving not only as the key amendment for existing sensors, but also one prerequisite to design any future CaM-based tools. We further speculate that for any exogenous entities containing endogenous proteins (or their key domains), protections should be well planned and implemented to constrict the actions of such factors in cells. Taking GCaMP and GCaMP-X as the example, the apoCaM contained in GCaMP should be under the control of the extra protection motif CBM to prevent the CaM motif from outreaching beyond the sensor. In this context, an alternative approach to design sensors and actuators might be appealing—totally de novo proteins/peptides for the particular cells/tissues/organisms through rational design or from other remote species in the phylogenetic tree. The former exemplars are computation-aided design of new calcium probes[55], and the successful cases of the latter are GFP[56] and ChR2[57], originated from ancient organisms jellyfish *Aequorea victoria* and alga *Chlamydomonas reinhardtii* respectively.

CaM (apoCaM or $Ca^{2+}$/CaM) is universally involved in complicated signaling in cells[19,58], e.g., CaM overexpression in neurons would severely perturb gene expressions in cortical neurons[44]. In addition to $Ca_V1$, various channels, receptors and enzymes could be perturbed by CaM-based probes[59–61], involving various physiological processes including membrane excitability, calcium dynamic, long-term potentiation or depression (LTP/LTD), synaptic transmission, CREB-dependent or other transcriptional signaling (MAPK/Erk, NFAT etc.), neural morphogenesis, cell cycles, cardiac hypertrophy and neural degeneration[9,62–65]. Can the principles derived from GCaMP-X

**Fig. 6** Schematic summary and sensor performance for GCaMP and GCaMP-X. **a** GCaMP defects and advantages of GCaMP-X. In neurons, GCaMP sensors behave as CaM-like proteins perturbing $Ca_V1$ gating and distorting $Ca^{2+}$ dynamics (1). The abnormal enhancement of $Ca_V1$ by cytosolic GCaMP underlies overgrown neurites. Different from endogenous CaM, GCaMP itself could no longer properly translocate into the nucleus in response to membrane excitation, which also dominant-negatively impairs the acute mobility of endogenous CaM. Once present in the nucleus (by accumulation or NLS), nuclear GCaMP perturbs transcription signaling and gene expression potentially through its aberrant CaM motif, impairing general health of neurons including $Ca^{2+}$ signals (reflected as abnormal sensor readouts) and neurite outgrowth (2). The new GCaMP-X sensors are designed with apoCaM protection and explicit localization (e.g., GCaMP-X_C for cytosolic $Ca^{2+}$) to eliminate GCaMP perturbations on $Ca_V1$ gating (3) and signaling (4). **b** Sensor performance of GCaMP-X_C validated by single-AP like $Ca^{2+}$ transients in HEK293 cells. Briefly, glass electrodes formed the Giga-Ohm seal with cell membrane (resting); then 3 ms ZAP for break-in generated a rapid $Ca^{2+}$ influx due to transient membrane rupture, immediately followed by a fast decay arising from strong $Ca^{2+}$ chelators of 10 mM BAPTA included in the pipette solution. Key indices of peak $\Delta F/F_0$, rise time $t_r$, decay time $t_d$ and SNR were compared between GCaMP6m and GCaMP6m-X_C. Epi-fluorescence images were acquired at the sampling rate of 100 Hz or higher. **c–e** Sensor performance of GCaMP-X_C validated with mechanosensing outer hair cells. Cells were transfected with GCaMP6m or GCaMP6m-X_C at P1 by electroporation, and cultured for 1 day (DIV 1) or 3 days (DIV 3). Fluid-jet pulses of 100, 300 or 500 ms were applied to cells (**c**). Representative images, normalized fluorescence changes ($\Delta F/F_0$) indicative of $Ca^{2+}$ dynamics and statistical summary of peak $\Delta F/F_0$ were compared between DIV 1 (**d**) and DIV 3 (**e**) cells expressing GCaMP6m or GCaM6m-X_C. Standard error of the mean (S.E.M.) and Student's $t$-test (two-tailed unpaired with criteria of significance: $*p < 0.05$; $**p < 0.01$, and $***p < 0.001$) were calculated when applicable, and n.s. denotes "not significant"

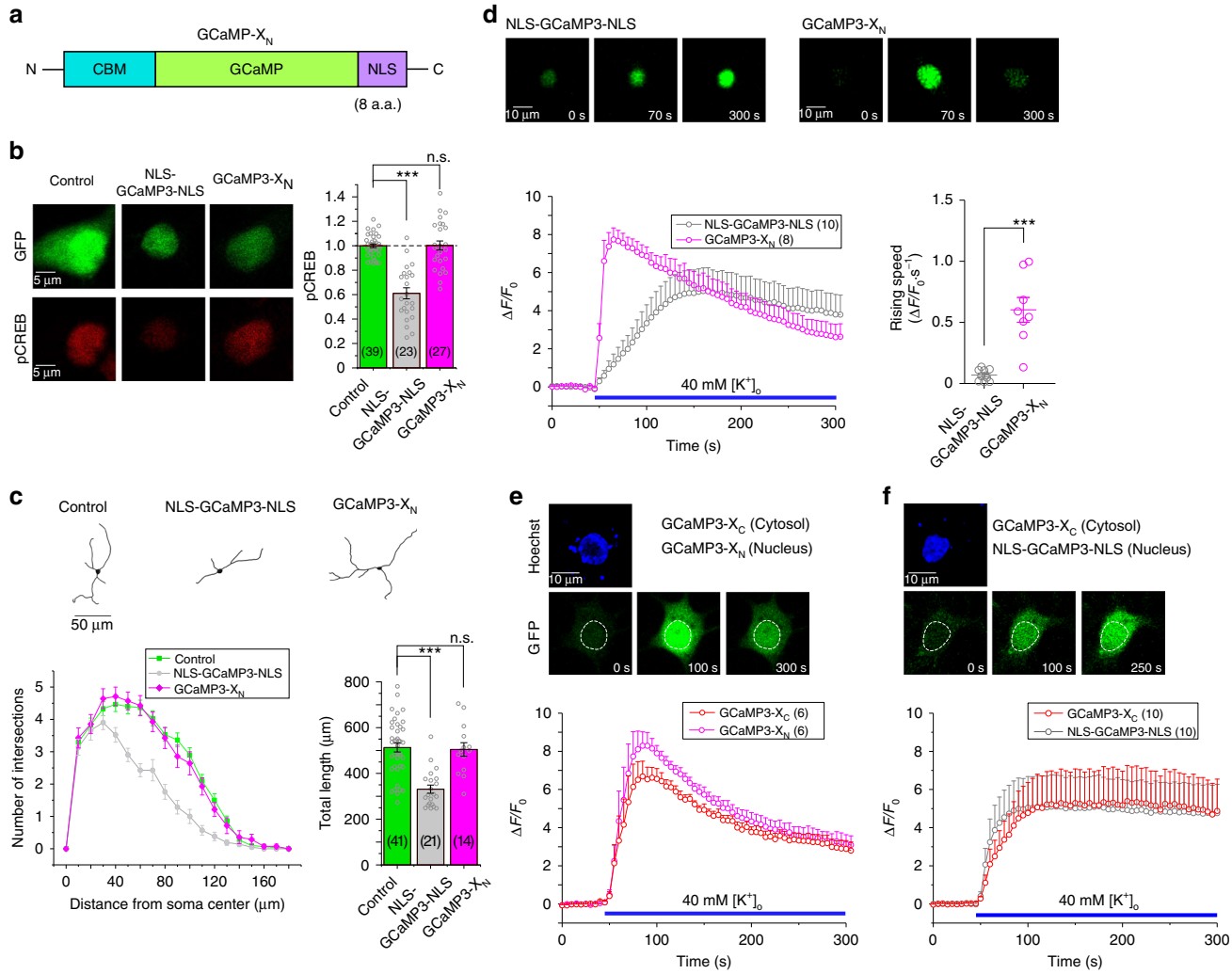

**Fig. 7** Characterizations and validations of GCaMP-$X_N$ targeting nuclear $Ca^{2+}$. **a** Design of GCaMP-$X_N$. Based on the design principle of GCaMP-X, similar to GCaMP-$X_C$, CBM was fused into N-terminus of GCaMP; and a nuclear localization signal (NLS) was tagged onto C-terminus of GCaMP. **b**, **c** Basic validations of GCaMP3-$X_N$ with pCREB signals and neurite outgrowth. Representative images of pCREB immunostaining (**b**, left), statistical summary of pCREB intensities (**b**, right), tracing of neurite morphology (**c**, upper), Sholl analysis and statistical summary of neurite length (**c**, lower) were compared among neurons transfected with YFP, NLS-GCaMP3-NLS or GCaMP3-$X_N$. **d** Different $Ca^{2+}$ dynamics resulted from neurons expressing NLS-GCaMP3-NLS or GCaMP3-$X_N$. Confocal images representing $Ca^{2+}$ fluorescence at three phases: before, during and at the end of extracellular stimuli of 40 mM $[K^+]_o$ (upper). $Ca^{2+}$ response ($\Delta F/F_0$) (lower left) and its rising speed ($\Delta F/F_0 \cdot s^{-1}$, normalized change of fluorescence per second, lower right) were averaged from multiple neurons (number indicated within parentheses), to compare NLS-GCaMP3-NLS with GCaMP3-$X_N$. **e**, **f** Simultaneous monitoring of cytosolic and nuclear $Ca^{2+}$ dynamics. Representative confocal images indicative of $Ca^{2+}$ fluorescence (upper in green) and time-dependent responses ($\Delta F/F_0$, lower) are to compare $Ca^{2+}$ dynamics in the cytosol vs. the nucleus of the same neuron upon 40 mM $[K^+]_o$ stimuli. Cytosolic $Ca^{2+}$ was monitored by GCaMP3-$X_C$ for both cases, whereas nuclear $Ca^{2+}$ was either by GCaMP3-$X_N$ (**e**) or by NLS-GCaMP3-NLS (**f**). Neurons were loaded with Hoechst 33342 to label the nuclei of neurons (upper, blue). Standard error of the mean (S.E.M.) and Student's $t$-test (two-tailed unpaired with criteria of significance: *$p < 0.05$; **$p < 0.01$, and ***$p < 0.001$) were calculated when applicable, and n.s. denotes "not significant"

design assure the exclusion of the above possible perturbations? According to this study, apoCaM interference is eliminated by CBM central to GCaMP-X design, indicative of an effective approach to avoid unwanted binding with apoCaM targets as well as their subsequent $Ca^{2+}/CaM$ binding/signaling. Other apoCaM-associated signaling events, e.g., of enzymes and cytoskeletal proteins, should also be effectively protected by our strategy[9]. Native protein targets of $Ca^{2+}/CaM$ binding should not be affected by either GCaMP or GCaMP-X, considering that the $Ca^{2+}/CaM$ motif preferentially binds onto the M13 motif within the same sensor because of their intramolecular closeness[11] and high affinity ($K_d \approx 4$ nM for $Ca^{2+}/CaM$)[37]. Therefore, in combination of CBM and M13, spatial restrictions are applied onto the CaM motif under all $Ca^{2+}$ conditions, thus effectively reducing its

overall mobility and concentration. Both apo and $Ca^{2+}$ forms of CaM or GCaMP-X are protected, greatly enhancing its resistance against any potential intermolecular binding with endogenous signaling proteins.

Due to compartmentalized nature of cellular calcium, a variety of organelle-targeted GECIs are developed, e.g., to address the questions specifically for nuclear calcium[34]. Nuclear presence of GCaMP has been considered as a sign of cell damages[5,7], also confirmed in this study by nucleus-accumulated GCaMP or by NLS-GCaMP-NLS. However, the reasons for such damages remain unclear. Instead of any direct interference with $Ca^{2+}/CaM$-binding proteins, it is more likely that GCaMP struck in the nucleus impairs certain critical apoCaM-related nuclear signaling thus eventually causing neural damages. Meanwhile, the

reason of nuclear invasion and aberrant accumulation has not been elucidated either. It is unlikely due to proteolytic cleavage as proposed[2], because nuclear GCaMP, although its readout was indeed attenuated, should mainly arise from neural damages and $Ca^{2+}$ distortion instead of any alteration in sensor functionalities (Fig. 7 and Supplementary Fig. 11). Generally, the size and shape of proteins may play a role in the passive entrance to the nucleus. However, based on the estimation and measurement, the slight difference in molecular weight between GCaMP ($\sim 50$ kDa)[66] vs. GCaMP-X ($\sim 54$ kDa) cannot account for their distinct cyto-nuclear localization. Also, according to the design of GCaMP-X and our homology modeling to compare their (apo) structures, the CBM binding to CaM constricts the freedom of the whole CaMP structure, making it less stretched out. Collectively, in this particular case of GCaMP-X vs. GCaMP, the size and shape are less likely to play any major roles. In addition, D3cpv has higher molecular weight ($\sim 73$ kDa), which is also a FRET sensor presumably in more "extended" shape, but still producing severe side-effects while getting accumulated in the nucleus ($\sim 40\%$ with the criteria of N/C ratio above 1, Supplementary Fig. 10). Based on these data and facts, we speculate that the nuclear accumulation of GCaMP might be caused by certain mechanisms related to activity-dependent or calcium-dependent cytonuclear translocation of CaM. Therefore, the bottom line is that GCaMP exerts its dominant negative effects through its CaM motif with incomplete and abnormal functionalities. As the roles and mechanisms related to nuclear $Ca^{2+}$ are not fully understood thus far, we expect GCaMP-$X_N$ to facilitate the progress of research in this direction, including the following key questions: what is the exact spatiotemporal relationship between nuclear and trans-membrane $Ca^{2+}$, and how exactly the information of activities is encoded into nuclear $Ca^{2+}$ and other related signals.

Based on above analyses, it is fairly possible that no or trivial effect could be induced when the actual expression level of GCaMP is extremely low (even lower than the fluorescence threshold in our imaging experiments). This is probably the case in some lines of GCaMP transgenic mice where no appreciable perturbation would be induced. In fact, much lower concentration of GCaMP has been reported in transgenic mice, e.g., GCaMP3/Ai38 mice ($\sim 5$ μM), compared to tens of μM or even higher GCaMP concentrations in viral infection[16]. In this context, we would not disagree with the claim from the researchers who routinely use GCaMP transgenic mice that these mice seem healthy and normal in general. Meanwhile, due to the above concerns we would like to suggest that cautions need to be taken when using the current lines of GCaMP transgenic mice, especially when suspicious complications were encountered, such as the epileptiform events from multiple GCaMP6 lines[14]. And GCaMP-X promises one direct solution with multi-fold advantages over conventional GCaMP. For instance, the low expression level required by GCaMP (to reduce severity of side-effects) constitutes one major drawback (low SNR due to insufficient GCaMP expression), potentially overcome by GCaMP-X with strong promoters to elevate expression and SNR in future lines.

As learned from this study, for any GECI, thorough tests should be carefully performed under various conditions beyond sensing characteristics. For GCaMP-X, it is imperative to validate its actual performance in vivo, especially for the goals of generating transgenic organisms. GCaMP-X would bring advantages of multi-fold to the new transgenic lines: free of complications and side-effects, allowing high expression levels, as well as excellent sensor characteristics including rapid kinetics and high SNR. Meanwhile, further optimization and more variants of GCaMP-X are expected, based on fast-growing structural and functional insights into GCaMP/CaM and related interactions of

both apo and $Ca^{2+}$ forms, to fulfill diverse and specific demands toward more precise and thorough information of cell $Ca^{2+}$.

## Methods

**Molecular biology.** $Ca_V 1.3$ $\alpha_{1D\_Long}$ (denoted as $\alpha_{1DL}$) variant (NM000720, Gen-Bank™ accession number), $Ca_V 2.2$ ($\alpha_{1B}$, NM_001243812.1), CaM and $CaM_{1234}$ were generously provided by Dr. David Yue (Johns Hopkins University). cDNA constructs of certain GCaMP variants were generous gifts from Drs. Minmin Luo and Sen Song (Tsinghua University). cDNA constructs encoding D3cpv (36323) and CaMPARI (60421) were purchased from Addgene. Inverse pericam was provided by Dr. Atsushi Miyawaki (Riken Brain Science Institute). $Ca_V 1.3$ $\alpha_{1D\_Short}$ (denoted as $\alpha_{1DS}$) variant was constructed by introducing a unique XbaI site following the IQ domain. YFP-tagged CaM/pcDNA3 was firstly generated by inserting YFP into pcDNA3 vector via unique KpnI and NotI sites, then CaM was fused to carboxyl termini of YFP via unique NotI and KpnI sites[22]. To target CFP and GCaMP3 to the nucleus of cells, a short NLS (PKKKRKV) was fused to both amino and carboxyl termini. NLS-CFP-NLS segment was amplified by PCR with flanking KpnI and XbaI then cloned directionally via these two unique sites into pcDNA3 expressing plasmids. NLS-GCaMP3-NLS segment was amplified by PCR with flanking BglII and NotI then cloned by replacing GCaMP3 via these two unique sites, yielding NLS-GCaMP3-NLS. IQ domain from neuromodulin (NM_017195.3) with modifications to render constitutive high-affinity apoCaM binding was amplified by PCR, with the sequence encoding MGMDE-LYKGTAATKIQAAFRGHITRKKLKDEKKGA (denoted as CBM). The fusion of CBM-GCaMP3 (GCaMP3-$X_O$) was ligated with EcoRI by overlap PCR with flanking BglII and NotI then cloned into pEGFP-N1 vector. CBM-GCaMP5G (GCaMP5G-$X_O$) and CBM-GCaMP6m (GCaMP6m-$X_O$) constructs were made by replacing GCaMP3 with appropriate PCR-amplified segments via unique EcoRI and NotI sites. To cytosolic localizations, NES (LALKLAGLDIGS) was fused to carboxyl terminus of GCaMP-$X_O$. Construct of CBM-GCaMP3-NES, CBM-GCaMP5G-NES and CBM-GCaMP6m-NES were named GCaMP3-$X_C$, GCaMP5G-$X_C$ and GCaMP6m-$X_C$, respectively. To specifically target the nucleus, NLS (PKKKRKV) was fused to its C-terminus of GCaMP-$X_O$. Construct of CBM-GCaMP3-NLS was named as GCaMP3-$X_N$. To target the membrane, Lck sequence of MGCGCSSNPEDDWMENIDVCENCHYPPPKLRIDPPDLAT was fused to the N-terminus of the sensors, then cloned into pcDNA3 vector via unique BamHI and NotI sites. The membrane-targeted versions of GCaMP3 and GCaMP3-X were denoted as GCaMP3$_M$ and GCaMP3-$X_M$, respectively.

TagRFP was amplified by PCR with flanking KpnI and NotI and cloned directionally via these two unique sites into pcDNA3 expressing plasmids. Then GCaMP3 was amplified with flanking NotI and XbaI and cloned via these two sites into pcDNA3, yielding TagRFP-GCaMP3 construct. To make $CaM_{1234}$P3 construct, $CaM_{1234}$ was amplified by overlap PCR with point mutation N60D and flanking MluI and NotI then cloned by replacing CaM N60D within GCaMP3. To delete M13 and cpEGFP from GCaMP3 and $GCaM_{1234}$P3, EGFP was amplified by PCR with flanking BglII and MluI then cloned by replacing M13 and cpEGFP via these two unique sites based on GCaMP3 and $GCaM_{1234}$P3, respectively. For chimeric $\alpha_{1DS}$-$DCT_F$, $\alpha_{1DS}$-$G_{12}$-GCaMP3 and $\alpha_{1DS}$-$G_{12}$-GCaMP3$\Delta$M13-cpEGFP, $\alpha_{1DS}$-$DCT_F$ was adopted from our previous work[22,67]. $G_{12}$ segment was PCR-amplified with SpeI and XbaI sites and inserted into $\alpha_{1DS}$ as template. GCaMP3 and GCaMP3$\Delta$M13-cpEGFP then were PCR-amplified with identical SpeI and XbaI sites and inserted into $\alpha_{1DS}$-$G_{12}$.

**Dissection and culturing of cortical neurons.** Cortical neurons were dissected from newborn ICR mice. Isolated tissues were digested with 0.25% trypsin for 15 min at 37 °C, followed by terminating the enzymatic reaction by DMEM supplemented with 10% FBS. The suspension of cells was sieved through a filter then centrifuged at 1000 rpm for 5 min. The cell pellet was resuspended in DMEM supplemented with 10% FBS and were plated on poly-D-lysine-coated 35 mm confocal dishes (In Vitro Scientific) or coverslips. After 4 h, neurons were maintained in Neurobasal medium supplemented with 2% B27, 1% glutaMAX-I (growth medium) for 5–6 days. Temperature should be around 37 °C with 5% $CO_2$ in the incubator. All animals were obtained from the laboratory animal research center, Tsinghua University. Procedures involving animals has been approved by local institutional ethical committees (IACUC, Tsinghua University).

**Transfection of cDNA constructs.** The HEK293 cell line (ATCC) used in this study was free of mycoplasma contamination, checked by PCR with primers 5′-GGCGAATGGGTGAGTAACACG-3′ and 5′-CGGATAACGCTTGCGA CCTATG -3′[22]. HEK293 cells were cultured in 60 mm dishes, and recombinant channels were transiently transfected according to an established calcium phosphate protocol[22]. We applied 5 μg of cDNA encoding the desired channel $\alpha_1$ subunit, along with 4 μg of rat brain $\beta_{2a}$ (M80545) and 4 μg of rat brain $\alpha_2\delta$ (NM012919.2) subunits. Additional 2 μg of cDNA was added as required in co-transfections. All of the above cDNA constructs were driven by a cytomegalovirus promoter. To enhance expression, cDNA for simian virus 40 T antigen (1–2 μg) was also co-transfected. Cells were washed with PBS 6–8 h after transfection and maintained in culture medium of supplemented DMEM, then incubated for at least 48 h in a water-saturated 5% $CO_2$ incubator at 37 °C before whole-cell recordings.

Neurons were cultured in 35 mm confocal dishes or coverslips, 2 μg of cDNA encoding the desired peptides were transiently transfected by Lipofectamine 2000 (Invitrogen). The opti-MEM containing plasmids and Lipofectamine 2000 was added to the Neurobasal medium for transfection. After 2 h, neurons were maintained in Neurobasal medium supplemented with 2% B27, 1% glutaMAX-I for 48 h.

**Infection of adeno-associated virus**. Virus of GCaMP6m (AAV-*Syn*-GCaMP6m), GCaMP6f (AAV-*Syn*-GCaMP6f) and GCaMP6m-X$_C$ (AAV-*Syn*-GCaMP6m-X$_C$) were provided by University of Pennsylvania Vector Core (USA) or Hanbio Biotechnology (China). Viruses were added to growth medium of the cortical neuron culture for 1–2 weeks.

**Whole-cell electrophysiology**. Whole-cell recordings of transfected HEK293 cells or cortical neurons were obtained at room temperature (25 °C) using an Axopatch 200B amplifier (Axon Instruments). Electrodes were pulled with borosilicate glass capillaries by a programmable puller (P-1000, Sutter Instruments) and heat-polished by a microforge (MF-830, Narishige), resulting in 1–3 MΩ resistances, before series resistance compensation of 70% or more. The internal solutions contained, (in mM): CsMeSO$_3$, 135; CsCl$_2$, 5; MgCl$_2$, 1; MgATP, 4; HEPES, 5; and EGTA, 5; at 290 mOsm adjusted with glucose and at pH 7.3 adjusted with CsOH. The extracellular solution contained (in mM): TEA-MeSO$_3$, 140; HEPES, 10; CaCl$_2$ or BaCl$_2$, 10; 300 mOsm, adjusted with glucose and at pH 7.3 adjusted with TEAOH, all according to the previous reports[22,67]. Whole-cell currents were generated from a family of step depolarizations ($-70$ to $+50$ mV from a holding potential of $-70$ mV) or a series of repeated step depolarizations ($-10$ mV from a holding potential of $-70$ mV). Currents were recorded at 2 kHz by the low-pass filter, and cells with Ca$^{2+}$ current amplitude less than 100 pA were not included in the analyses. Traces were acquired at a minimum repetition interval of 30 s. P/8 leak subtraction was used throughout.

For patch-clamp recordings on neurons, isolated cortical neurons were cultured for 11 days in order to obtain larger calcium current. To isolate Ca$_V$1.3 currents, Ca$_V$1.2, N- and P/Q-type Ca$^{2+}$ channel currents were blocked by pre-incubating neurons in Tyrode's solution containing 1 μM nimodipine (Sigma-Aldrich), 1 μM ω-conotoxin GVIA (Sigma-Aldrich, or Alomone Labs), 1 μM ω-conotoxin MVIIC (Sigma-Aldrich, or Alomone Labs) for 30 min before recording, according to cocktail recipes[65,68,69], validated with our recordings in which Ca$_V$1.3 was the dominant component (~80%), compared to Ca$_V$2.3 (~16%) and Ca$_V$1.2 (contributing to the rest). Neurons were used within 1 h after pre-incubation and recorded in bath solution with blocker cocktail.

In all electrophysiological experiments, recordings from at least four cells were analyzed ($n \geq 4$).

**Immunocytochemistry**. Cortical neurons which were transfected on 5th day and used on 7th day, were silenced with 1 μM TTX for 24 h to prevent AP, then transferred to 40 mM [K$^+$]$_o$ solution (1 μM TTX, 95 mM NaCl, 40 mM KCl, 1 mM MgCl$_2$, 15 mM HEPES, 1.2 mM CaCl$_2$, at 300 mOsm adjusted with glucose) or 20 mM [K$^+$]$_o$ solution (1 μM TTX, 115 mM NaCl, 20 mM KCl, 1 mM MgCl$_2$, 15 mM HEPES, 1.2 mM CaCl$_2$, at 300 mOsm adjusted with glucose) and treated for 30 min (detecting pCREB) or 5 min (detecting CaM). Then cortical neurons were rinsed briefly in phosphate-buffered saline (PBS), fixed with ice cold 4% paraformaldehyde in PBS (pH 7.4) for 20 min at the room temperature (~25 °C), then washed three times with ice-cold PBS. Fixed cells were then permeabilized with 0.3% Triton X-100 and blocked with 10% normal goat serum in PBS for 1 h at room temperature, and incubated overnight at 4 °C in primary antibodies of pCREB (Rabbit mAb #9198, Cell Signaling Technology, Species Cross-Reactivity: Human, Mouse, Rat, Dilutions: 1:1000), CREB (Rabbit mAb #9197, Cell Signaling Technology, Species Cross-Reactivity: Human, Mouse, Rat, Monkey, D. melanogaster, Dilutions: 1:500) or CaM (Rabbit mAb #5197-1, Epitomics, Species Cross-Reactivity: Human, Mouse, Rat, Dilutions: 1:500). The next day, cells were washed with PBS three times, incubated at room temperature for 2 h in 1:800 dilution of anti-rabbit-Alexa633 (Invitrogen), and washed with PBS three times (incubated with Hoechst 33342 for 5 min when nuclear counterstain was needed). Cells were imaged with ZEISS Laser Scanning Confocal Microscope (LSM710) (Carl Zeiss) and ZEN 2009 software. Fluorescence intensity was quantified and analyzed with ImageJ (NIH). Calculations of nuclear fluorescence intensity were aided with nuclear regions stained by Hoechst 33342. In all immunocytochemistry experiments, images of at least ten cells were analyzed ($n \geq 10$).

**Apoptosis assay**. Cortical neurons on confocal dishes were incubated with fluorescein-labelled Annexin V reagent (Keygen Biotech, 1:100 dilution) for 15 min at the room temperature (~25 °C). Cells were washed once with Tyrode's solution and observed with a confocal laser microscope (LSM710).

**Fluorescence Ca$^{2+}$ imaging with GCaMP or GCaMP-X**. Cortical neurons were pre-incubated in 5 mM [K$^+$]$_o$ solution (130 mM NaCl, 5 mM KCl, 1 mM MgCl$_2$, 15 mM HEPES, 2 mM CaCl$_2$, at 300 mOsm adjusted with glucose) and perfused with 20 or 40 mM [K$^+$]$_o$ solution, then washed out by 5 mM [K$^+$]$_o$. Epi-fluorescent imaging experiments were performed with an inverted fluorescence microscope

(Ti-U, Nikon, Japan) and Neo sCMOS camera (Andor Technology). The light source was from the mercury lamp filtered at appropriate wavelengths for GFP by the optical filters mounted at the computer-controlled filter wheel (Sutter Instrument) for excitation, subsequently passing the dichroic mirror and the emission filters. Operations and measurements were controlled by the iQ software (Andor Technology). For confocal Ca$^{2+}$ imaging, a similar protocol was followed with ZEISS Laser Scanning Confocal Microscope (LSM710) (Carl Zeiss) and ZEN 2009 software. Fluorescence intensity ($F$) was subtracted from its background, to calculate the index of $\Delta F/F_0$, where $F_0$ is the baseline fluorescence averaged from 1 s or three or more data points at rest, and $\Delta F = F - F_0$.

**Fluorescence Ca$^{2+}$ imaging with Fura-2**. Cortical neurons were cultured on coverslips and were loaded in Tyrode's solution with 5 μM Fura-2 AM (Abcam) and 0.02% Pluronic F-127 (Invitrogen) in incubator for 30 min. Neurons were pre-incubated in 5 mM [K$^+$]$_o$ solution and perfused with high [K$^+$]$_o$ solution, then washed out by 5 mM [K$^+$]$_o$. Fluorescence ratio ($F_{340}/F_{380}$) was achieved with a standard ratiometric Ca$^{2+}$ imaging system, which included an inverted IX71epi-fluorescence microscope (Olympus), a camera of EMCCD DU-897D (Andor Technology), the light source of Lambda DG-4 (Sutter instrument), the excitation filters of 340 nm and 380 nm: FF01-340/26-25 (Semrock) and FF01-380/14-25 (Semrock), the emission filters of ~510 nm: FF01-504/12-25 (Semrock), and the dichromatic mirror: DM 400 in U-MWU2 cube (Olympus). Images were acquired and analyzed by MetaFluor software (Molecular Devices).

**Induction of Ca$^{2+}$ dynamics in HEK293 cells**. To generate Ca$^{2+}$ waveforms resembling that of single AP, the glass electrode for whole-cell patch clamp was filled with the intracellular solution containing strong Ca$^{2+}$ chelators of 10 mM BATPA. Subsequent to the standard GΩ-seal formation, 3 ms ZAP stimulation from the amplifier (Axopatch 200B) was applied for membrane break-in. Due to transient membrane rupture, a rapid Ca$^{2+}$ influx was induced (to mimic the rising phase of AP-associated Ca$^{2+}$). Immediately after, whole-cell configuration was stably established, and BAPTA quickly diffused into the cell, which altogether attenuated intracellular Ca$^{2+}$ (mimicking the decay phase of single-AP Ca$^{2+}$ waveform). An alternative approach to produce rapid Ca$^{2+}$ influx was by way of voltage-gated Ca$_V$2.2 channels overexpressed in HEK293 cells (Ca$_V$2.2 current amplitudes were within the range of 0.5–2 nA). A 100 ms of $+10$ mV voltage step was applied to depolarize the membrane and activate Ca$_V$2.2 channels. Under this approach, Ca$^{2+}$ influx was more controllable since Ca$_V$2.2 currents ($I_{Ca}$) were simultaneously monitored by whole-cell patch clamp while GCaMP fluorescence images were acquired at 100 Hz or higher. Another stimulation method was also used to generate Ca$^{2+}$ dynamics by acetylcholine (Ach) activation of endogenous mus-carinic receptors in HEK293 cells[5]. Bath containing 10 μM Ach (TargetMol) was briefly applied to cells and then washed out. Rise time $t_r$ was defined as the time (from the start of stimulation) to reach the half of peak $\Delta F/F_0$. Decay time $t_d$ was defined as the time (from the time of peak) to fall back to the half of peak $\Delta F/F_0$. SNR was calculated by the ratio of peak $\Delta F/F_0$ over standard derivation (SD) of baseline signals (1 s duration).

**Ca$^{2+}$ imaging with outer hair cells**. According to previously published protocol[39], the organ of Corti containing outer hair cells was isolated from P1 mice. For electroporation, glass electrodes (2 μM diameter) were used to deliver plasmids (1 μg/μl in 1 × HBSS) to outer hair cells. A series of three pulses was applied at 1 s intervals with a magnitude of 60 V and duration of 15 ms (ECM 830 square wave electroporator, BTX). Organ was cultured for 1–3 days in DMEM/F12 medium with 10% FBS and 1.5 μg/ml ampicillins. For Ca$^{2+}$ imaging, outer hair cells were imaged on an upright Olympus BX51WI microscope. Hair bundles of outer hair cells were stimulated with a fluid jet applied through a glass electrode filled with bath solution (1.3 mM CaCl$_2$, 144 mM NaCl, 0.7 mM NaH$_2$PO$_4$, 5.8 mM KCl, 0.9 mM MgCl$_2$, 5.6 mM glucose, and 10 mM H-HEPES, pH 7.4). Stimuli were controlled by Patchmaster 2.35 software (HEKA) and 20 psi air pressure were applied for each stimulation. Sampling rate of images was at 2 s, and duration of fluid-jet stimulation was increased from 100 to 300 ms, and then to 500 ms.

**Image analysis of neurite morphology**. Measurement of the length and Sholl analysis for neurites were performed with Imaris 7.7.2 (Bitplane). Only non-overlapping neurons were selected for analysis and images of at least 14 neurons were analyzed ($n \geq 14$).

**Data analysis and statistics**. Data were analyzed in Matlab, GraphPad Prism and OriginPro software. Standard error of the mean (S.E.M.) and Student's *t*-test (two-tailed with criteria of significance: *$p < 0.05$; **$p < 0.01$, and ***$p < 0.001$) were calculated when applicable, and n.s. denotes "not significant".

**Data availability**. The authors declare that all data supporting the findings of this study are available within this article and the Supplementary Information File; or available from the corresponding author upon request.

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

## Acknowledgements

We thank all Liu-Lab (X-Lab) members for discussions and help, and particularly Dr. M. Liu for her pilot experiments on GCaMP effects on Ca$_V$1.3 gating. We acknowledge the researchers who shared constructs as indicated in the Methods section. We also thank Drs. K.X. Yuan, J. Yao and Y.C. Jia, and graduate students X.J. Piao and C.G. Chen all from Tsinghua University, for providing technical help. This work is supported by Natural Science Foundation of China (NSFC) grants 21778034, 31370822, 81171382, 81371604, 31571080 and 31522025 and Beijing Natural Science Foundation (BNSF) grant 7142089.

## Author contributions

Y.Y., Y.H., N.L., Y.L., and L.G. performed the experiments and analyses. Y.Y. and N.L. participated in manuscript preparations including text and figures. L.Z., S.S., and W.X. contributed to live tissue/animal experiments and analysis. X.L. conceived the project, designed the experiments and wrote the paper.

## Additional information

**Competing interests:** The authors declare no competing interests.

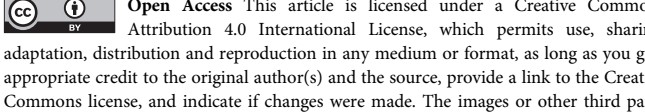

