## [Peer Review File · Nature Communications]

Reviewer #1 (Remarks to the Author):

Yang et al. have attempted to uncover the mechanism responsible for aberrant subcellular localization and performance in the GCaMP series of genetically encodable calcium ion indicators (GECIs). The authors have identified the Cav1 channel as being responsible for unwanted GCaMP-dependent effects on cell physiology, they have characterized the interaction, and have even provided a solution in the form of GCaMP-X, all of which will be highly beneficial to the biological and neuroscience communities. The solution presented in this manuscript (adding an apoCaM binding peptide to the N-terminus of existing GCaMP sensors), is new and creative and seems to provide a significant reduction of previously described problems with this class of sensors. Specifically, the GCaMP-X series restores normal excitation-transcription signalling by preventing GCaMP-apoCaM and GCaMP-Ca²⁺/CaM from significantly interacting with the CaV1 distal carboxyl tail region while consequently restoring Ca²⁺ dependent inactivation of the channel. This effect was characterized with electrophysiological characterization of the channel in HEK cells and cultured cortical neurons where localization, intracellular calcium concentration, and CREB activity were restored to native levels.

Overall, I find this work to be compelling and insightful and potentially suitable for publication in Nature Communications. This work is likely to be controversial and generate a lot of discussion in the field of calcium imaging, particularly in the area of neuroscience where GCaMP is most heavily used. On one side you will have workers who have themselves noticed artefacts when using GCaMP and will be excited to see both an explanation and a solution. On the other side will be workers who routinely use GCaMP transgenic mice that seem healthy and normal, and will have trouble accepting that there could be any major perturbations of cell physiology associated with expression of GCaMP.

For the most part, the authors have provided the results and discussion that will convince the workers in the first camp. I feel that a bit more needs to be done to convince the skeptical workers in the second camp. Specifically, the authors need to clearly reconcile their results with the existence of healthy transgenic mice. They touch on this in the final paragraph but I did not find this to be a satisfying discussion. I think that the point they are trying to make is that the existing mice are largely free of phenotypical artefacts due to low expression levels of GCaMP. This is a very reasonable explanation, and something that needs to be emphasized consistently through the manuscript. For figure 1g and 3b, the neurite length and pCREB levels, respectively, should also be plotted versus total GCaMP fluorescence intensity (to represent expression level). These could be included as Supplementary figures. I suspect that these new plots will show that the artefacts are directly correlated with expression level, perhaps with a correlation coefficient that is at least as good as the one when they are plotted against N/C ratio. Assuming that the analysis shows that lower concentration gives decreased artefacts, and that the authors agree that transgenic mice are normal due to low expression levels, one of the most important advantages of GCaMP-X is that it will allow the creation of mice with stronger promoters driving GCaMP-X expression. Higher expression would result in mice with brighter cells which should give higher S/N Ca²⁺ imaging.

When reporting a new sensor, it is typical to fully characterize its characteristics (i.e., extinction coefficient, quantum yield, sensitivity, dissociation constant, and in cellulo brightness). The authors have not done this, and have rather made the reasonable claim that fusing another protein to the N-terminus is unlikely to affect the performance. I feel this is reasonable for most characteristics of the GCaMP protein, with the exception of response kinetics. Due to the interaction of CaM with CBM, it is likely that the kinetics of the sensor have been perturbed. The authors should perform cell based experiments to compare the kinetics of GCaMP-X to the corresponding sensors without the CBM domain.

I list below a number of additional issues to be addressed.

1) The claim that nuclear accumulation of GCaMP “clearly evidenced from long-term expressions of GCaMP” does not seem to be supported by the Zariwala et al., 2012 reference. Specifically, long term transgenic expression does not result in nuclear accumulation, but AAV-mediated expression did result in nuclear accumulation. This result suggests that it is not a property of GCaMP itself that leads to nuclear accumulation. The authors need to address and discuss this result which seems to contradict their hypothesis.

2) Figure 1h and elsewhere: I am not an expert in statistics, but it seems to me that the authors should be reporting (and showing error bars for) standard deviations rather than standard errors of the mean. For something like ‘total neurite length’, it is the variation in the lengths between many different cells that is relevant here, and the standard deviation should be used accordingly. Also, it is the standard deviation which is used in the Student’s t-test, which the authors are using to determine the statistical significance of their results.

3) The references for supplementary figure 1 do not match the manuscript.

4) The N/C ratio for EGFP control cells should be provided.

5) Figure 1c, the EGFP labelling does not make a compelling argument for localization of the reporter. Possibly using a longer exposure time and saturating the soma would help visualize the dendrites.

6) Figure 1e, “base” should read “based”. Furthermore, there is no mention of the time point after transfection of this figure.

7) Page 6, “neurite outgrowth seemed being” should read “neurite growth seemed to be promoted in neurons...”

8) Page 7, the word ‘damaged’ here is too vague of a description.

9) Figure 2, there are statistics for GCaMP6 but none for GCaMP3. The use of GCaMP3 and GCaMP6 does not seem to follow a pattern and this is a recurring issue throughout the manuscript. Also, Figure 2b the y-axis text is scaled wrong (pA/pF).

10) Page 21: It is not clear to me what ‘Camgo’ is. Also, the phrase “alien species” does not seem appropriate.

11) Page 22: The Roda, 2010 reference seems like a poor choice as a citation for GFP, considering all of the great papers and reviews that were authored by the Nobel prize winners.

12) In the discussion section, the authors should refine their discussion of other indicators a bit, to take into account that CaM-based indicators have been engineered to not interact with endogenous targets. They should also mention the use of troponin-based indicators which would obviously not suffer from the problems detailed here. Finally, there are now many different colors of GCaMP-type indicators available and the authors should make it clear that these problems are likely to persist with these other color variants.

13) The proposed reasons for the translocation to the nucleus are not very compelling. It is not at all clear to me why the N/C ratio should differ between cells in an otherwise homogeneous population (unless it is all just a matter of expression level, as discussed above). Additional speculation on possible mechanisms might be appropriate. For example, I wonder if there could be a post-translational modification of GCaMP that is occurring in some cells but not others, and this is causing translocation.

Reviewer #2 (Remarks to the Author):

The thorough and extensive analysis by Yang et al provides important information about side effects of the widely used genetically encoded Ca sensors of the GCaMP family. A major effect is interference with Ca signaling to the nucleus and CREB-mediated transcriptional control. The authors find that the calmodulin moiety of GCaMP3 and 6 affects Ca-dependent inactivation of L-type Ca channels. This is somewhat a surprise because it has not been considered so far but retrospectively it is quite logical because Calmodulin (CaM) in its Ca-free state (apoCaM) has to pre-associate with L-type channels for proper Ca-dependent inactivation (CDI). If CDI is impaired increased Ca influx will positively or negatively influence downstream effects such as CREB-mediated gene expression.

A second important aspect of this new work is the establishment of a modified GCaMP, which the authors called GCaMPX, which carries a binding site for apoCaM so that GCaMP6X will not pre-associate with L-type channels and thereby perturb their CDI.

I only have a few Minor Concern and would leave it up to the authors if they would want to address those before potential publication:

1. On page 10, authors state 'Normalized pCREB intensity in neurons expressing GCaMP3 exhibited a negative correlation with N/C ratio with GCaMP.' Please indicate how pCREB intensity was normalized. In addition it would be desirable to show antibody staining for total CREB in Fig. 3a.

2. In Fig 4e, authors show that GCaMP3 does not translocate to the nucleus in response to high KCl, unlike endogenous CaM. Under this condition (GCaMP3 overexpression), how does endogenous CaM behave upon high KCl treatment? If endogenous CaM moves to the nucleus upon high KCl, what would be the reason of altered transcription?

3. Figure 4h: it would be desirable to show Fura-2 imaging from control cells that have not been transfected with a sensor at all

First of all, we are very grateful for the efforts and help from you and the reviewers for our manuscript NCOMMS-17-17029, entitled “GCaMP-X prevents its calmodulin from perturbing calcium channel-dependent excitation-transcription coupling”. By providing additional data and analyses, we have revised the manuscript in response to the insightful and constructive comments, point by point as outlined below (reviewers’ comments are in ***Bold Italic*** and our responses are in normal fonts; Page and Line numbers are denoted by **P** and **L**).

Reviewer #1

Yang et al. have attempted to uncover the mechanism responsible for aberrant subcellular localization and performance in the GCaMP series of genetically encodable calcium ion indicators (GECIs). The authors have identified the Cav1 channel as being responsible for unwanted GCaMP-dependent effects on cell physiology, they have characterized the interaction, and have even provided a solution in the form of GCaMP-X, all of which will be highly beneficial to the biological and neuroscience communities. The solution presented in this manuscript (adding an apoCaM binding peptide to the N-terminus of existing GCaMP sensors), is new and creative and seems to provide a significant reduction of previously described problems with this class of sensors. Specifically, the GCaMP-X series restores normal excitation-transcription

signalling by preventing GCaMP-apoCaM and GCaMP-Ca²⁺/CaM from significantly interacting with the CaV1 distal carboxyl tail region while consequently restoring Ca²⁺ dependent inactivation of the channel. This effect was characterized with electrophysiological characterization of the channel in HEK cells and cultured cortical neurons where localization, intracellular calcium concentration, and CREB activity were restored to native levels.

Overall, I find this work to be compelling and insightful and potentially suitable for publication in Nature Communications. This work is likely to be controversial and generate a lot of discussion in the field of calcium imaging, particularly in the area of neuroscience where GCaMP is most heavily used. On one side you will have workers who have themselves noticed artefacts when using GCaMP and will be excited to see both an explanation and a solution. On the other side will be workers who routinely use GCaMP transgenic mice that seem healthy and normal, and will have trouble accepting that there could be any major perturbations of cell physiology associated with expression of GCaMP.

For the most part, the authors have provided the results and discussion that will convince the workers in the first camp. I feel that a bit more needs to be done to convince the skeptical workers in the second camp. Specifically, the authors need to clearly reconcile their results with the existence of healthy transgenic mice. They touch on this in the final paragraph but I did not find this to be a satisfying discussion. I think that the point they are trying to make is that the existing mice are largely free of phenotypical artefacts due to low expression levels of GCaMP. This is a very reasonable explanation, and something that needs to be emphasized consistently through the manuscript. For figure 1g and 3b, the neurite length and pCREB levels, respectively, should also be plotted versus total GCaMP fluorescence intensity (to represent expression level). These could be included as Supplementary figures. I suspect that these new plots will show that the artefacts are directly correlated with expression level, perhaps with a correlation coefficient that is at least as good as the one when they are plotted against N/C ratio. Assuming that the analysis shows that lower concentration gives decreased artefacts, and that the authors agree that transgenic mice are normal due to low expression levels, one of the most important advantages of GCaMP-X is that it will allow the creation of mice with stronger promoters driving GCaMP-X expression. Higher expression would result in mice with brighter cells which should give higher S/N Ca²⁺ imaging.

We agree with the reviewer on the above viewpoints, especially the major point that the artefacts and side-effects of GCaMP are highly correlated with the extent of overexpression. We performed additional experiments to quantify the correlation between the expression level of GCaMP and the indices of neurite length (**Supplementary Fig. 3**) and pCREB level (**Supplementary Fig. 5a**). As

predicated by the reviewer, the side-effects of GCaMP are strongly correlated with its expression level. The (linear) correlation coefficients are of high (absolute) values: 0.7 (length) and 0.6 (pCREB), comparable to those indexed with N/C ratio: 0.7 (**Fig. 1g**) and 0.8 (**Fig. 3b**) respectively.

We also appreciate the insights of the reviewer regarding the expression level of GCaMP including the potential benefits of GCaMP-X over conventional GCaMP in transgenic mice. Accordingly, we have revised the last part of the manuscript, appending additional discussion (**P25 – 26**). We agree with the reviewer on most of the comments. Meanwhile, we would like to point out that the enhancement of neurite length and pCREB by cytosolic GCaMP, although just moderate compared to the control group (YFP only), has been consistently observed throughout this study. In fact, CaM overexpression in neurons would perturb gene expressions in cortical neurons (Pang *et al.* J. Neuroscience 2010); and gain of function in firing rates was observed from hippocampal neurons of GCaMP5G transgenic mice (Gee *et al.* Neuron 2014). And recently, multiple lines of GCaMP6 mice are reported to have major abnormalities in their brain activity (Steinmetz *et al.* eNeuro 2017). Also, GCaMP expression induced hypertrophy/cardiomegaly in the heart of GCaMP2 transgenic mice (Tallini *et al.* PNAS 2006), resembling CaM overexpression in ventricular myocytes (Colomer *et al.* Endocrinology 2004). Nevertheless, perturbations of the above kind have not been widely reported in neurons or animals thus far, probably due to the following reasons. First, the downregulation of neurite outgrowth and pCREB is linked to neurite or cell death, more apparent and problematic than the upregulation, the latter of which may only be reflected into less discernible phenotypes in the nervous or other systems. Second, evaluation of GCaMP level largely relies on the basal fluorescence of Ca²⁺-free GCaMP (to exclude negative neurons expressing no probe), which is far less bright than GFP or Ca²⁺/GCaMP. Thus, for the neurons categorized as “low expression/fluorescence” in our experiments may still have substantial GCaMP being overexpressed, underlying the upregulation of neurite growth and CREB signals. Therefore, it is fairly possible that no or trivial effect could be induced when the actual expression level of GCaMP is extremely low (even lower than what we observed here), which is probably the case in some popular lines of GCaMP transgenic mice. In this context, ultralow expression is required for current transgenic mice, which, as the reviewer pointed out, is the major drawback (low SNR) potentially to be overcome by GCaMP-X with strong promoters (enhanced expression and SNR) in future lines.

In all, we would not disagree with the claim from the researchers who routinely use GCaMP transgenic mice that these mice seem healthy and normal in general. Meanwhile, due to the above concerns we would like to suggest that cautions need to be taken when using the current lines of GCaMP transgenic mice, especially when suspicious complications (*e.g.*, epileptiform events in

GCaMP6 mice, Steinmetz *et al.* eNeuro 2017) are encountered. And GCaMP-X promises one of the potential solutions with multi-fold advantages.

When reporting a new sensor, it is typical to fully characterize its characteristics (i.e., extinction coefficient, quantum yield, sensitivity, dissociation constant, and in cellulo brightness). The authors have not done this, and have rather made the reasonable claim that fusing another protein to the N-terminus is unlikely to affect the performance. I feel this is reasonable for most characteristics of the GCaMP protein, with the exception of response kinetics. Due to the interaction of CaM with CBM, it is likely that the kinetics of the sensor have been perturbed. The authors should perform cell based experiments to compare the kinetics of GCaMP-X to the corresponding sensors without the CBM domain.

As suggested by the reviewer, we have examined the kinetics of GCaMP6 vs GCaMP6-X. The approach we employed was to induce comparable Ca^{2+} dynamics mimicking that of one single action potential (AP), by fast break-in with whole-cell patch-clamp and ZAP stimulus. In the bath with abundant Ca^{2+} (10 mM), a standard patch pipette containing strong Ca^{2+} buffer of 10 mM BAPTA sealed onto the membrane of HEK cell. Upon ZAP break-in, a transient fluorescence increase was introduced by brief influx of extracellular Ca^{2+} due to membrane rupture and reseal, followed by a fast decrease due to Ca^{2+} chelating by BAPTA. Such transient Ca^{2+} produced similar fluorescence signals as measured by GCaMP6 vs GCaMP6-X, indexed with peak $\Delta F/F_0$, SNR, rise time t_r and decay time t_d (**Fig. 6b** and **Supplementary Fig. 12**). GCaMP6m and GCaMP6m-X exhibited statistically insignificant difference, if there was any, in their kinetic parameters (rise time and decay time), which were also comparable to single AP-driven Ca^{2+} in neurons (Chen *et al.* Nature 2013).

We didn't directly compare these sensors in neurons, because of Ca^{2+} distortions and overall perturbations in neurons overexpressing conventional GCaMP. The experimental design as we devised above should provide more fair and precise comparison of the sensing performance between the conventional GCaMP6 and the new GCaMP-X.

We reason the least perturbation of Ca^{2+} sensing (or Ca^{2+} -CaM binding) within GCaMP-X should be largely due to the special feature of CBM, which is genetically engineered to tightly bind the CaM motif at rest ($K_d \approx 2 \mu\text{M}$) (Black *et al.* Biochemistry 2006; Chapman *et al.* JBC 1991). When Ca^{2+} rises, the M13 motif (Blumenthal *et al.* PNAS 1985) has much higher affinity with Ca^{2+} /CaM ($K_d \approx 4 \text{ nM}$) than that between CBM and Ca^{2+} /CaM ($K_d = 2 - 40 \mu\text{M}$ depending on whether the four binding sites of CaM are fully bound with Ca^{2+}). So the function of CBM is confined to apoCaM or ultralow Ca^{2+} conditions, relieving the concern that the introduction of this CBM motif might interfere with Ca^{2+} sensing mechanisms originally implemented in GCaMP.

We have revised the manuscript accordingly (P16 L20 – P17 L6; P12 L16 – 22).

I list below a number of additional issues to be addressed.

1) The claim that nuclear accumulation of GCaMP “clearly evidenced from long-term expressions of GCaMP” does not seem to be supported by the Zariwala et al., 2012 reference. Specifically, long term transgenic expression does not result in nuclear accumulation, but AAV-mediated expression did result in nuclear accumulation. This result suggests that it is not a property of GCaMP itself that leads to nuclear accumulation. The authors need to address and discuss this result which seems to contradict their hypothesis.

We agree with the reviewer on the study comparing transgenic mice with virus infection (Zariwala et al. 2012 J. Neuroscience), where the authors claim that long term transgenic expression does not result in nuclear accumulation. For the original text regarding nuclear accumulation “clearly evidenced from long-term expressions of GCaMP” (now please refer to P5 L16), we meant gene delivery methods of virus infection or plasmid transfection exhibiting clear nuclear accumulation (e.g., Figure 2 C-E in Zariwala’s study). Meanwhile, regarding another study of transgenic mice (Madisen et al. Neuron 2015) mentioned in Discussion, we did claim that “In comparison with viral infection, nucleus-filling neurons from GCaMP6 transgenic mice are much less, but still discernible” (P22 L1). This is in part based on Figure 4G of Madison’s study, where the authors reported “no or **little** invasion of transgene proteins with time”. From their confocal images, we did see some neurons with clear nuclear filling of GCaMP. Unprotected GCaMP in the nucleus causes neural damages and cell apoptosis, and thus any nuclear invasion of GCaMP should be considered as a red alert, which is much relieved for GCaMP-X even when present in nuclei.

In general, the side-effects including nuclear filling are much less often noticed in transgenic mice. Based on the analyses suggested by the reviewer, nuclear filling and expression level of GCaMP are highly correlated. Transgenic expression tailored by different promoters could constrain GCaMP level within its ultralow range, as one workaround solution for GCaMP perturbations.

Meanwhile, we cannot exclude the possibility of “a property of GCaMP itself that leads to nuclear accumulation”. In fact, GCaMP-X₀ without any localization signal, even when abundantly expressing in cells, no longer accumulated in the nucleus to perturb pCREB signals (Fig. 5a, b), suggesting apoCaM protection is sufficient to prevent nuclear filling. Is it the overexpression of GCaMP that directly leads to nuclear accumulation? If nuclear accumulation of GCaMP can cause cell damages, it is more likely that some other processes involving apoCaM (GCaMP) actually lead to aberrant nuclear invasion in the first place (so the damages are subsequent to nuclear invasion). Side effects directly by NLS-tagged GCaMP support this hypothesis. In this context, we speculate that properties

intrinsic to GCaMP (instead of simply its high expression level) should trigger its nuclear accumulation, which would be well manifested when GCaMP reaches high (thus more severe perturbations on Cav1 in neurons). Because of these prerequisite conditions for clear manifestation, we agree with the reviewer that “it is not a property of GCaMP itself that leads to nuclear accumulation”. Indeed, in HEK cells (lacking the machinery of Cav1-dependent E-T coupling), no difference between GCaMP and GCaMP-X has been found under various testing conditions, distinct from native neurons.

To avoid the potential confusions regarding transgenic vs non-transgenic expression of GCaMP as pointed out, we have revised the relevant paragraphs (P21 – 22).

2) Figure 1h and elsewhere: I am not an expert in statistics, but it seems to me that the authors should be reporting (and showing error bars for) standard deviations rather than standard errors of the mean. For something like ‘total neurite length’, it is the variation in the lengths between many different cells that is relevant here, and the standard deviation should be used accordingly. Also, it is the standard deviation which is used in the Student’s t-test, which the authors are using to determine the statistical significance of their results.

In response to this comment, we double-checked published studies in related research fields. And it is common, as we found out, that the results using mean \pm SEM are presented in bar graphs along with statistical significance (Student’s *t*-test). For instance, dendritic length was depicted in bar graph by mean \pm SEM (Krey *et al.* Nature Neuroscience 2013; Gomez-Ospina *et al.* Cell 2006), and the latter study also employed Student’s *t*-test to calculate *p* values. We here would like to point out:

- 1) Both SD (standard deviation) and SEM (standard error of the mean) could be used in reporting statistical results.
- 2) SEM is commonly used, emphasizing on the (un)certainty of the population mean; whereas SD is to describe how close the samples are.
- 3) Mathematically, the two have direct relationship: $SEM=SD/SQRT(N)$, where N is the sample size.

By “total neurite length”, we meant to estimate the mean neurite length per cell for neurons from each group; SEM here was used to describe how well the mean value was estimated for each group, *e.g.*, if enough number of neurons were sampled and measured, etc. Meanwhile, “the variation in the lengths between many different cells” can also provide important information, for which we revised all the bar graphs throughout this manuscript by adding the original data points as one way to depict the variations among individual neurons, *e.g.*, **Fig. 1h**.

3) The references for supplementary figure 1 do not match the manuscript.

We have revised the main text (**P3 L11**) referring to Supplementary Fig. 1, to ensure the citations matching the list of literature: Nakai *et al.* 2001, Tallini *et al.* 2006, Tian *et al.* 2009, Akerboom *et al.* 2012, Chen *et al.* 2013.

4) The N/C ratio for EGFP control cells should be provided.

We have appended the trend of N/C ratio for EGFP control in **Fig. 1a**, which turned to be a stable line as expected.

5) Figure 1c, the EGFP labelling does not make a compelling argument for localization of the reporter. Possibly using a longer exposure time and saturating the soma would help visualize the dendrites.

We have revised **Fig. 1c** to better visualize the dendrites. As advised, the fluorescence images have been improved with more clear morphology. Also, following the style in **Fig. 1f** and other figures, we appended the images tracing the branches for each neuron (**Fig. 1c**, top row in black and white). In both cases, wide-field images were routinely taken along with fluorescence images for each view, to provide confirmations when necessary.

6) Figure 1e, “base” should read “based”. Furthermore, there is no mention of the time point after transfection of this figure.

The mistake has been corrected in the caption of **Fig. 1e**. And a protocol was added following the style of **Fig. 1a**, to indicate the time point (two days) after transfection when the results were obtained.

7) Page 6, “neurite outgrowth seemed being” should read “neurite growth seemed to be promoted in neurons...”

We have revised it accordingly (**P6 L19**).

8) Page 7, the word ‘damaged’ here is too vague of a description.

We have revised the text according to this comment. Now it reads:

“First, as often noticed and reported in GCaMP applications, we here explicitly demonstrated that nuclear GCaMP, as one red alert sign along with the time or level of GCaMP expression, caused detrimental effects on neurons leading to neurite or cell loss.” (**P7 L3 – 6**)

9) Figure 2, there are statistics for GCaMP6 but none for GCaMP3. The use of GCaMP3 and GCaMP6 does not seem to follow a pattern and this is a recurring issue throughout the manuscript. Also, Figure 2b the y-axis text is scaled wrong (pA/pF).

We have taken care of this issue in the following aspects:

- 1) Unified the styles for all the electrophysiological data into two main categories: recombinant channels or neuronal currents. For recombinant channels, we showed the voltage dependent profiles of inactivation and activation indexed with S_{Ca} and J_{Ca} values, *e.g.*, **Fig. 2c, d** and **Fig. 4b, c**, where significance tests were also appended.
- 2) For neurons, I_{Ca} recordings were shown at the single step of -10 mV, and S_{Ca} and J_{Ca} were compared in bar graphs explicitly, *e.g.*, **Fig. 2b** and **Fig. 4d**.
- 3) Moved the data of GCaMP6 effects on recombinant channels (originally in **Fig. 4**) into **Supplementary Fig. 9** for clarity and simplicity (since GCaMP3 data already provided the major evidence, details see below).

We would like to clarify that the different presenting styles are not due to any discrimination between GCaMP3 and GCaMP6 (in fact, both produced similar effects in almost all aspects), but rather about different experimental settings. The advantages of recombinant channels in HEK cells ensure larger amplitudes, less contamination, more robustness, etc., altogether providing more thorough and precise characterizations than native currents. So in general, recombinant data and analyses would be more fully and thoroughly available than neuronal recordings as seen in **Fig. 2** and **Fig. 4**. In response to this comment, in addition the above two items, we also paid attention to the issue of “following a pattern” or the presenting style, and have revised the figures when necessary, including **Fig. 2, Fig. 4, Supplementary Fig. 4** and **Supplementary Fig. 8-10**.

Also, in the manuscript, to avoid potential artifact arising from the different size of cells, we routinely used current density (pA/pF) instead of current amplitude (pA) for both neuronal and recombinant channels, *e.g.*, **Fig. 2**.

10) Page 21: It is not clear to me what ‘Camgoo’ is. Also, the phrase “alien species” does not seem appropriate.

We have corrected the typo to Camgaroos (Yu *et al.* J of Neuroscience 2003) (**P22 L13**). And we have replaced the phrase “alien species” with “other remote species in the phylogenetic tree” (**P23 L4**).

11) Page 22: The Roda, 2010 reference seems like a poor choice as a citation for GFP, considering all of the great papers and reviews that were authored by the Nobel prize winners.

We have replaced the citation of GFP with another widely-cited review article by Dr. RY Tsen (Tsen *Annu Rev Biochem* 1998) (P23 L5).

12) In the discussion section, the authors should refine their discussion of other indicators a bit, to take into account that CaM-based indicators have been engineered to not interact with endogenous targets. They should also mention the use of troponin-based indicators which would obviously not suffer from the problems detailed here. Finally, there are now many different colors of GCaMP-type indicators available and the authors should make it clear that these problems are likely to persist with these other color variants.

We agree with the reviewers that other CaM-based indicators may or may not cause side effects in cells since each specific design should be examined experimentally in a case-by-case basis. However, to our knowledge, potential interferences with apoCaM targets have not been given enough attention until this study. Without intentionally taking care of this apoCaM problem, CaM-based sensors are at high risk to produce similar artifacts and aversive effects in neurons. In fact, for the limited set of sensors we tested (D3cpv, CaMPARI and inverse pericam), all exhibited more or less perturbations on Cav1 gating and signaling (**Supplementary Fig. 10**).

For non-CaM based sensors, such as the troponin-based indicators of TN-XL (Mank *et al. Biophysics J.* 2006), we agree with the reviewer that they are unlikely to suffer from the problems detailed here. Meanwhile, we agree that these problems are likely to persist for other color variants of GCaMP, such as RCaMP (Inoue *et al. Nature Methods* 2015).

We have revised the text accordingly (P22 L10 - L17).

13) The proposed reasons for the translocation to the nucleus are not very compelling. It is not at all clear to me why the N/C ratio should differ between cells in an otherwise homogeneous population (unless it is all just a matter of expression level, as discussed above). Additional speculation on possible mechanisms might be appropriate. For example, I wonder if there could be a post-translational modification of GCaMP that is occurring in some cells but not others, and this is causing translocation.

We agree that for GCaMP the level of expression is highly correlated with nuclear invasion and accumulation. However, it is unlikely that the nuclear translocation is simply or directly caused by high expression, since we demonstrated that GCaMP-X with apoCaM protection no longer accumulated in the nucleus even with high levels or extended time of expression in neurons (**Fig. 5a**). As we speculated, the potential factor underlying this aberrant nuclear accumulation should be related to activity-dependent or calcium-dependent cytonuclear translocation of CaM, one critical

event in excitation-transcription coupling. By immunostaining we examined the effects of cytosolic GCaMP on the level of nuclear CaM in response to high K⁺ stimuli to neurons. In contrary to CaM translocation into the nucleus as observed from control neurons (**Fig. 3d**), neither GCaMP and CaM levels gained any significant increase, strongly suggesting that GCaMP exerted dominant negative effects on normal CaM translocation (**Supplementary Fig. 6d**, and please refer to item 2 from reviewer #2 for more details). In line with these data, it seems to us that the correlation between GCaMP level and nuclear accumulation reflects the overall perturbations on Cav1 and general health of neurons. Such perturbations would be added up when GCaMP level is high, eventually leading to massive nuclear invasion and accumulation which would further exaggerate GCaMP damages.

We also agree with the reviewer that one possibility could be post-translational modification, such as proteolysis potentially linked to its altered sensing performance (Tian *et al.* Nature Methods 2009). Our data thus far did not provide direct support to such linkage, as in **Fig. 7f**, where nuclear GCaMP, although its readout was indeed attenuated, should be due to neural damages and distorted cellular Ca²⁺ instead of any change in sensor characteristics. If examined in HEK cells, GCaMP and GCaMP-X did not result into any difference in Ca²⁺ sensing (**Supplementary Fig. 11**), further supporting the Cav1-centered mechanisms we proposed in this study.

We have revised the text accordingly (**P21 L7 – 18**).

Reviewer #2 (Remarks to the Author):

The thorough and extensive analysis by Yang et al provides important information about side effects of the widely used genetically encoded Ca sensors of the GCaMP family. A major effect is interference with Ca signaling to the nucleus and CREB-mediated transcriptional control. The authors find that the calmodulin moiety of GCaMP3 and 6 affects Ca-dependent inactivation of L-type Ca channels. This is somewhat a surprise because it has not been considered so far but retrospectively it is quite logical because Calmodulin (CaM) in its Ca-free state (apoCaM) has to pre-associate with L-type channels for proper Ca-dependent inactivation (CDI). If CDI is impaired increased Ca influx will positively or negatively influence downstream effects such as CREB-mediated gene expression.

A second important aspect of this new work is the establishment of a modified GCaMP, which the authors called GCaMPX, which carries a binding site for apoCaM so that GCaMP6X will not pre-associate with L-type channels and thereby perturb their CDI.

I only have a few Minor Concern and would leave it up to the authors if they would want to address those before potential publication:

1. On page 10, authors state 'Normalized pCREB intensity in neurons expressing GCaMP3 exhibited a negative correlation with N/C ratio with GCaMP.' Please indicate how pCREB intensity was normalized. In addition it would be desirable to show antibody staining for total CREB in Fig. 3a.

The mean pCREB intensity of control neurons under basal conditions was set as the standard value, based on which each neuron under test was normalized to achieve 'normalized pCREB intensity' for all experimental groups.

As suggested, we have conducted the immunostaining experiments with CREB-antibody for **Fig. 3a**, where pCREB levels were examined previously. In **Supplementary Fig. S5b**, the (total) CREB level stayed unchanged, indicating no correlation between GCaMP expression/distribution and CREB level (correlation coefficient was as low as 0.06).

We have revised the relevant text (**P10 L14 – 22**).

2. In Fig 4e, authors show that GCaMP3 does not translocate to the nucleus in response to high KCl, unlike endogenous CaM. Under this condition (GCaMP3 overexpression), how does endogenous CaM behave upon high KCl treatment? If endogenous CaM moves to the nucleus upon high KCl, what would be the reason of altered transcription?

We have conducted the experiment to examine the level of nuclear CaM immunostaining in response to acute application of high KCl in GCaMP-overexpressing neurons. In **Supplementary Fig. 6**, data suggest that the acute translocation of endogenous CaM was severely impaired in GCaMP-overexpressing neurons.

For neurons expressing GCaMP (in the cytosol), the physiological responses to membrane excitation, *e.g.*, elevation of nuclear CaM level and phosphorylation of CREB, are strongly attenuated. In other words, GCaMP produces dominant negative effects on acute CaM translocation (**Supplementary Fig. 6d**) and activity-dependent transcription signals (**Supplementary Fig. 6c**) in response to high K⁺ stimuli. Reportedly, upon excitation CaM in close vicinity to membrane channels of Cav1 would be recruited for cytonuclear translocation (Ma *et al.* Cell 2014). And for GCaMP-overexpressing neurons, as we reasoned, such acute mobilization of CaM should be much less efficient or severely hindered since Cav1 channels are now surrounded mostly by aberrant CaM

(immobile GCaMP) instead of endogenous WT CaM. GCaMP enrichment (over endogenous CaM) local to Cav1 has been proved by its strong perturbations on Cav1/CaM gating.

We have revised the relevant text (**P11 L20 – P12 L1**).

3. Figure 4h: it would be desirable to show Fura-2 imaging from control cells that have not been transfected with a sensor at all

We understand that it would be desirable to show Fura-2 imaging results from control neurons as the ‘standard’ Ca²⁺ dynamics for comparison. However, sensors based on direct Ca²⁺-binding have the intrinsic issue of Ca²⁺-buffering which affects the ‘real’ Ca²⁺ signal to different extents depending on concentrations, binding affinities and other properties of the probes (Helmchen *et al.* Biophysical J. 1996; Rose *et al.* Frontiers in Molecular Neuroscience, 2014). For the same kind of probes (based on the same mechanisms of action/binding), relatively fair comparison can be made by controlling the concentration of the probes, as we did for GCaMP-X vs GCaMP (similar levels of expression). However, for distinct types of probes such as GCaMP vs Fura-2, due to their fundamental differences, it is very likely that they would result into different read-outs even all the other conditions are the same (*e.g.*, the same probe concentration and the same Ca²⁺ signal). Nevertheless, on top of Fig. 4f, we appended Fura-2 data of two different concentrations (5 μM and 20 μM) in control neurons, shown below:

As expected, higher concentration of 20 μM Fura-2 caused more attenuation on cellular Ca²⁺ signals compared with 5 μM Fura-2. For neurons loaded with 5 μM Fura-2, adding GCaMP or GCaMP-X further attenuated the signal. However, more importantly, GCaMP resulted into larger Ca²⁺ signal (than GCaMP-X), due to its abnormal enhancement of Ca²⁺ currents via Cav1 channels.

We have revised the text according to the above results and analyses (**P13 L18 – P14 L2**).

In **Fig. 4f** (previous Fig. 4h), we focused on the particular problem of GCaMP and our solution of GCaMP-X, overlooking the details on buffering effects common to many Ca^{2+} sensors. We thank the reviewer for giving us the chance to clarify these matters.

Most of the above revisions have been highlighted (in yellow) in the main manuscript and the Supplementary Figures file. For your convenience, I also attach the list of all the references cited in this response letter. Thanks again for your considering our work and we look forward to hearing back from you soon.

Sincerely,

Xiaodong Liu, Ph.D.

References

- Akerboom, J., Chen, T.W., Wardill, T.J., Tian, L., Marvin, J.S., Mutlu, S., Calderon, N.C., Esposti, F., Borghuis, B.G., Sun, X.R., *et al.* (2012). Optimization of a GCaMP Calcium Indicator for Neural Activity Imaging. *Journal of Neuroscience* *32*, 13819-13840.
- Black, D.J., Leonard, J., and Persechini, A. (2006). Biphasic Ca²⁺-dependent switching in a calmodulin-IQ domain complex. *Biochemistry* *45*, 6987-6995.
- Blumenthal, D.K., Takio, K., Edelman, A.M., Charbonneau, H., Titani, K., Walsh, K.A., and Krebs, E.G. (1985). Identification of the calmodulin-binding domain of skeletal muscle myosin light chain kinase. *Proceedings of the National Academy of Sciences of the United States of America* *82*, 3187-3191.
- Chapman, E.R., Au, D., Alexander, K.A., Nicolson, T.A., and Storm, D.R. (1991). Characterization of the calmodulin binding domain of neuromodulin. Functional significance of serine 41 and phenylalanine 42. *J Biol Chem* *266*, 207-213.
- Chen, T.-W., Wardill, T.J., Sun, Y., Pulver, S.R., Renninger, S.L., Baohan, A., Schreiter, E.R., Kerr, R.A., Orger, M.B., Jayaraman, V., *et al.* (2013). Ultrasensitive fluorescent proteins for imaging neuronal activity. *Nature* *499*, 295-300.
- Colomer, J.M., Terasawa, M., and Means, A.R. (2004). Targeted expression of calmodulin increases ventricular cardiomyocyte proliferation and deoxyribonucleic acid synthesis during mouse development. *Endocrinology* *145*, 1356-1366.
- Gee, J.M., Smith, N.A., Fernandez, F.R., Economo, M.N., Brunert, D., Rothermel, M., Morris, S.C., Talbot, A., Palumbos, S., Ichida, J.M., *et al.* (2014). Imaging activity in neurons and glia with a Polr2a-based and cre-dependent GCaMP5G-IRES-tdTomato reporter mouse. *Neuron* *83*, 1058-1072.
- Gomez-Ospina, N., Tsuruta, F., Barreto-Chang, O., Hu, L., and Dolmetsch, R. (2006). The C terminus of the L-type voltage-gated calcium channel Ca(V)1.2 encodes a transcription factor. *Cell* *127*, 591-606.
- Helmchen, F., Imoto, K., and Sakmann, B. (1996). Ca²⁺ buffering and action potential-evoked Ca²⁺ signaling in dendrites of pyramidal neurons. *Biophys J* *70*, 1069-1081.
- Inoue, M., Takeuchi, A., Horigane, S., Ohkura, M., Gengyo-Ando, K., Fujii, H., Kamijo, S., Takemoto-Kimura, S., Kano, M., Nakai, J., *et al.* (2015). Rational design of a high-affinity, fast, red calcium indicator R-CaMP2. *Nat Methods* *12*, 64-70.
- Krey, J.F., Pasca, S.P., Shcheglovitov, A., Yazawa, M., Schwemberger, R., Rasmusson, R., and Dolmetsch, R.E. (2013). Timothy syndrome is associated with activity-dependent dendritic retraction in rodent and human neurons. *Nat Neurosci* *16*, 201-209.
- Madisen, L., Garner, A.R., Shimaoka, D., Chuong, A.S., Klapoetke, N.C., Li, L., van der Bourg, A., Niino, Y., Egolf, L., Monetti, C., *et al.* (2015). Transgenic mice for intersectional targeting of neural sensors and effectors with high specificity and performance. *Neuron* *85*, 942-958.
- Mank, M., Reiff, D.F., Heim, N., Friedrich, M.W., Borst, A., and Griesbeck, O. (2006). A FRET-based calcium biosensor with fast signal kinetics and high fluorescence change. *Biophys J* *90*, 1790-1796.
- Nakai, J., Ohkura, M., and Imoto, K. (2001). A high signal-to-noise Ca²⁺ probe composed of a single green fluorescent protein. *Nature Biotechnology* *19*, 137-141.

- Pang, Z.P., Xu, W., Cao, P., and Südhof, T.C. (2010). Calmodulin suppresses synaptotagmin-2 transcription in cortical neurons. *The Journal of biological chemistry* 285, 33930-33939.
- Rose, T., Goltstein, P.M., Portugues, R., and Griesbeck, O. (2014). Putting a finishing touch on GECIs. *Front Mol Neurosci* 7, 88.
- Seidemann, E., Chen, Y., Bai, Y., Chen, S.C., Mehta, P., Kajs, B.L., Geisler, W.S., and Zemelman, B.V. (2016). Calcium imaging with genetically encoded indicators in behaving primates. *Elife* 5.
- Tallini, Y.N., Ohkura, M., Choi, B.R., Ji, G., Imoto, K., Doran, R., Lee, J., Plan, P., Wilson, J., Xin, H.B., *et al.* (2006). Imaging cellular signals in the heart in vivo: Cardiac expression of the high-signal Ca²⁺ indicator GCaMP2. *Proceedings of the National Academy of Sciences* 103, 4753-4758.
- Tian, L., Hires, S.A., Mao, T., Huber, D., Chiappe, M.E., Chalasani, S.H., Petreanu, L., Akerboom, J., McKinney, S.A., Schreiter, E.R., *et al.* (2009). Imaging neural activity in worms, flies and mice with improved GCaMP calcium indicators. *Nature Methods* 6, 875-881.
- Tsien, R.Y. (1998). The green fluorescent protein. *Annu Rev Biochem* 67, 509-544.
- Yu, D., Baird, G.S., Tsien, R.Y., and Davis, R.L. (2003). Detection of calcium transients in *Drosophila* mushroom body neurons with camgaroo reporters. *J Neurosci* 23, 64-72.
- Zariwala, H.A., Borghuis, B.G., Hoogland, T.M., Madisen, L., Tian, L., De Zeeuw, C.I., Zeng, H., Looger, L.L., Svoboda, K., and Chen, T.W. (2012). A Cre-Dependent GCaMP3 Reporter Mouse for Neuronal Imaging In Vivo. *Journal of Neuroscience* 32, 3131-3141.

Reviewer #1 (Remarks to the Author):

My first review of this manuscript was quite positive and I concluded that this work is “compelling and insightful and potentially suitable for publication in Nature Communications”. This view has not changed, and I anticipate that this work will be subject to quite a bit of post-publication scrutiny due to its interesting, important, and possibly controversial claims. The authors have done a good job of addressing the comments in the first round of review. Many of my original comments and suggestions were attempts to reconcile the observations reported in this work, with the day-to-day experience of workers who use GCaMP constructs and mice. Along this same line, many workers who express fluorescent protein constructs in cells will be familiar with the importance of molecular size (and shape) in determining whether proteins are able to passively enter the nucleus or not. Depending on the shape of the protein (that is, globular like GCaMP or extended like a FRET construct) there is a transition point somewhere in the 50-100 kDa range where proteins can no longer passively enter the nucleus. In the main text, the authors should provide the MW of both GCaMP and GCaMP-X, and mention that the extra additional size is likely playing a role in preventing GCaMP-X variants from entering the nucleus. The increased size may be more relevant here than CaV1/CaM signaling. Accordingly, this sentence, *“GCaMP-XO itself already very much relieved the abnormal nuclear accumulation, suggesting that the unknown reason for abnormal GCaMP nuclear accumulation is closely related to CaV1/CaM signaling in normal cytonuclear translocation”*, should be modified accordingly. This sentence *“Moreover, GCaMP-X without any localization signal, even abundantly expressing in cells, no longer accumulated in the nucleus to perturb pCREB signals (Fig. 5a, b), suggesting apoCaM protection is sufficient to prevent nuclear filling”*, may also need to be reworded.

Once the authors have made this suggested change, the manuscript will be suitable for publication in Nature Communications.

Reviewer #2 (Remarks to the Author):

The authors have thoroughly addressed the issues raised in the Critique. Especially important is the recent article by Steinmetz et al., 2017, as now cited. Accordingly, GCaMP transgenic mice often show neuronal phenotypes. The work by Liu and colleagues provides a molecular explanation and at the same time a solution to this issue. I recommend immediate publication for the benefit of the research community.

Re: Decision on manuscript NCOMMS-17-17029A

Reviewers' comments are in ***Bold Italic*** and our responses are in normal fonts and Page and Line numbers are denoted by P and L respectively.

Reviewer #1 (Remarks to the Author):

My first review of this manuscript was quite positive and I concluded that this work is "compelling and insightful and potentially suitable for publication in Nature Communications". This view has not changed, and I anticipate that this work will be subject to quite a bit of post-publication scrutiny due to its interesting, important, and possibly controversial claims. The authors have done a good job of addressing the comments in the first round of review. Many of my original comments and suggestions were attempts to reconcile the observations reported in this work, with the day-to-day experience of workers who use GCaMP constructs and mice. Along this same line, many workers who express fluorescent protein constructs in cells will be familiar with the importance of molecular size (and shape) in determining whether proteins are able to passively enter the nucleus or not. Depending on the shape of the protein (that is, globular like GCaMP or extended like a FRET construct) there is a transition point somewhere in the 50-100 kDa range where proteins can no longer passively enter the nucleus. In the main text, the authors should provide the MW of both GCaMP and GCaMP-X, and mention that the extra additional size is likely playing a role in preventing GCaMP-X variants from entering the nucleus. The increased size may be more relevant here than CaV1/CaM signaling. Accordingly, this sentence, "GCaMP-XO itself already very much relieved the abnormal nuclear accumulation, suggesting that the unknown reason for abnormal GCaMP nuclear accumulation is closely related to CaV1/CaM signaling in normal cytonuclear translocation", should be modified accordingly. This sentence "Moreover, GCaMP-X without any localization signal, even abundantly expressing in cells, no longer accumulated in the nucleus to perturb pCREB signals (Fig. 5a, b), suggesting apoCaM protection is sufficient to prevent nuclear filling", may also need to be reworded.

Once the authors have made this suggested change, the manuscript will be suitable for publication in Nature Communications.

We thank the reviewer again for the positive review and the support of publication. We agree with the reviewer that the general importance of molecular size (and shape) in determining whether proteins are able to passively enter the nucleus or not. To address the concern for this particular case, we estimated and compared the MW for the sensor proteins tested in this work. GCaMP-X (~54 kDa) in fact is only slightly heavier than

GCaMP (~50 kDa, also see Akerboom et al. 2009), so unlikely to account for their drastic difference in cytonuclear localization. Also, the reviewer raised an interesting viewpoint that the more “extended shape” may “play a role in preventing GCaMP-X variants from entering the nucleus”. However, according to the design of GCaMP-X and our homology modeling to compare their (apo) structures, the CBM binding to CaM constricts the freedom of the whole GCaMP structure, making it less stretched out. Collectively, we agree with the reviewer that generally the additional size and the shape of the protein would play important roles in nuclear entrance; but in this particular case of GCaMP-X vs GCaMP, these factors are less likely to play any major roles. As additional support, we noticed that in the list of sensors we tested, D3cpv has higher MW (~73 kDa), which is also a FRET sensor potentially in more “extended” shape, but still producing severe side-effects while getting accumulated in the nucleus (~40% with the criteria of N/C ratio above 1, Supplementary Fig. 10).

We have modified the text according to the suggestions and above considerations (P14 L1-5; P21 L7-9; P24 L6-15). The first sentence of “GCaMP-X₀ itself...” is changed to:

“GCaMP-X₀ itself already very much relieved the abnormal nuclear accumulation. The reason responsible for abnormal GCaMP nuclear accumulation is not very clear yet. Basic properties of the protein itself may play a role, such as the size and shape. Or alternatively, such abnormality could be due to some unexpected interference with normal Ca_v1-dependent cytonuclear translocation processes.”

And the second sentence of “Moreover...” is changed to:

“Moreover, GCaMP-X without any localization signal, even with high expression levels, no longer accumulated in the nucleus to perturb pCREB signals (Fig. 5a, b), suggesting apoCaM protection is an effective way to alleviate nuclear filling.”

Reviewer #2 (Remarks to the Author):

The authors have thoroughly addressed the issues raised in the Critique. Especially

important is the recent article by Steinmetz et al., 2017, as now cited. Accordingly, GCaMP transgenic mice often show neuronal phenotypes. The work by Liu and colleagues provides a molecular explanation and at the same time a solution to this issue. I recommend immediate publication for the benefit of the research community.

We appreciate the reviewer's support and recommendation to publish our work.